# Gene age shapes the transcriptional landscape of sexual morphogenesis in mushroom-forming fungi (Agaricomycetes)

Zsolt Merényi[1], Máté Virágh[1], Emile Gluck-Thaler[2], Jason C Slot[3], Brigitta Kiss[1], Torda Varga[1], András Geösel[4], Botond Hegedüs[1], Balázs Bálint[1], László G Nagy[1,5]*

[1]Synthetic and Systems Biology Unit, Biological Research Center, Szeged, Hungary; [2]Department of Biology, University of Pennsylvania, Philadelphia, United States; [3]College of Food, Agricultural, and Environmental Sciences, Department of Plant Pathology, Ohio State University, Columbus, United States; [4]Institute of Horticultural Science, Department of Vegetable and Mushroom Growing, Hungarian University of Agriculture and Life Sciences, Budapest, Hungary; [5]Department of Plant Anatomy, Institute of Biology, Eötvös Loránd University, Budapest, Hungary

**\*For correspondence:**
lnagy@fungenomelab.com

**Competing interest:** The authors declare that no competing interests exist.

**Abstract** Multicellularity has been one of the most important innovations in the history of life. The role of gene regulatory changes in driving transitions to multicellularity is being increasingly recognized; however, factors influencing gene expression patterns are poorly known in many clades. Here, we compared the developmental transcriptomes of complex multicellular fruiting bodies of eight Agaricomycetes and *Cryptococcus neoformans*, a closely related human pathogen with a simple morphology. In-depth analysis in *Pleurotus ostreatus* revealed that allele-specific expression, natural antisense transcripts, and developmental gene expression, but not RNA editing or a 'developmental hourglass,' act in concert to shape its transcriptome during fruiting body development. We found that transcriptional patterns of genes strongly depend on their evolutionary ages. Young genes showed more developmental and allele-specific expression variation, possibly because of weaker evolutionary constraint, suggestive of nonadaptive expression variance in fruiting bodies. These results prompted us to define a set of conserved genes specifically regulated only during complex morphogenesis by excluding young genes and accounting for deeply conserved ones shared with species showing simple sexual development. Analysis of the resulting gene set revealed evolutionary and functional associations with complex multicellularity, which allowed us to speculate they are involved in complex multicellular morphogenesis of mushroom fruiting bodies.

## Editor's evaluation

This study sought to systematically identify key aspects of the transcriptional landscape in fungi that exhibit complex multicellularity (CM), associated with fruiting bodies. The authors examined a series of parameters of expression signatures, concluding that the best predictor of a gene behavior in the CM transcriptome was evolutionary age. Thus, the expression pattern of fruiting bodies showed a distinct gene age-related stratification, where it was possible to sort out genes related to general sexual processes from those likely linked to morphogenetic aspects of the CM fruiting bodies. Notably, these results do not support a developmental hourglass concept, which is the rather predominant hypothesis in metazoans, including some analysis in fungi.

## Introduction

The emergence of multicellularity has been one of the most influential transitions in evolution (*Knoll, 2011*; *Smith and Szathmary, 1995*). However, while simple multicellular aggregations evolved several times and evidence is accumulating that these transitions may not have had as many genetic obstacles as originally thought (*Abedin and King, 2008*; *Kiss et al., 2019*; *Nagy et al., 2018*; *Sebé-Pedrós et al., 2017*), origins of complex multicellularity (CM) seem to be rare evolutionary events. Simple multicellularity refers to cell aggregations, colonies, or filaments, whereas CM comprises three-dimensional (3D) organisms in which not all cells are in direct contact with the environment. CM probably required the evolution of mechanisms for transport, cell adhesion, and complex developmental programs (*Knoll, 2011*). Diverse studies suggest that, besides the changes in gene content or protein sequence, the evolution of gene expression and genome regulation is also important in the transition to CM (*King et al., 2003*; *Merényi et al., 2020*; *Sebé-Pedrós et al., 2018*).

Uniquely across life on Earth, fungi show evidence for multiple evolutionary origins of CM (*Nagy, 2018*; *Nguyen et al., 2017*). CM in fungi, as defined by *Knoll, 2011*, refers to fruiting bodies and some other 3D structures (e.g., sclerotia, ectomycorrhizae; see *Nagy et al., 2018*). CM in fungi is restricted to certain stages of the life cycle and starts by the transition from simple hyphal growth to 3D organization, for example, during the development of sexual fruiting bodies. This allows real-time transcriptomic readouts of changes associated with transitions in complexity level, which make fungi an ideal model system to investigate CM. Fungi reach the highest level of multicellular complexity in fruiting bodies of Agaricomycetes (*Kües and Navarro-González, 2015*; *Nagy, 2018*), which includes most industrially cultivated edible and medicinal mushrooms. CM fruiting bodies in the Agaricomycetes have been widely studied by transcriptomic approaches; however, the interpretation of transcriptomes has been complicated by the lack of an understanding of the general principles of transcriptome evolution. This has, among other factors, impeded the definition of core CM- and development-related genes and pathways and thus reaching a general synthesis on the genetics of CM in the Agaricomycetes. Recent studies of fruiting body development reported species-specific and conserved genes (*Krizsán et al., 2019*; *Nguyen et al., 2017*), natural antisense transcripts (NATs) (*Muraguchi et al., 2015*; *Ohm et al., 2010*; *Shao et al., 2017*), allele-specific expression (ASE) (*Gehrmann et al., 2018*), RNA editing (*Zhu et al., 2014*), small RNA (*Lau et al., 2018*), alternative splicing (*Krizsán et al., 2019*), chromatin remodeling (*Vonk and Ohm, 2021*), as well as developmental hourglass (*Cheng et al., 2015*); however, how widespread these are during and how significant their contributions to fruiting body development are not known.

Similarly, several genes and cellular processes have been identified in agaricomycete fruiting bodies. Fruiting bodies are composite structures in which structural cell types enclose reproductive ones (basidia, meiospores) into a protective environment. Basidium and spore development are evolutionarily significantly older than CM fruiting bodies (*Virágh et al., 2021*). The genes underlying basidium and spore development show up in developmental transcriptomes and, if not properly accounted for, can blur signals for real CM-related genes. Accordingly, while some hitherto identified genes can be linked to CM functions (e.g., defense of fruiting bodies; see *Künzler, 2018*), most fruiting body-expressed genes, including those related to cell wall remodeling (*Liu et al., 2021*), transcriptional regulation, selective protein degradation (*Krizsán et al., 2019*), or complex secretomes (*Almási et al., 2019*), could relate either to CM or more general functions.

One of the main goals of this study was to systematically tease apart the components and driving forces of transcriptome evolution in a CM fungus. To this end, we examined NATs, ASE, and RNA editing in a well-resolved developmental transcriptome of *Pleurotus ostreatus* (oyster mushroom). We found that developmental expression and ASE of a gene strongly correlate with the gene's evolutionary age. Building on this observation, the second aim of this study was to identify conserved gene families whose expression patterns associate with CM in the Agaricomycetes. For this, we compared the transcriptomes of eight CM fungi and that of a species with simple sexual development (*Cryptococcus neoformans*). The gene age-related stratification of developmental transcriptomes was prevalent across all examined species; however, these were not compatible with developmental hourglass concept as postulated for animals. Nevertheless, the evolutionary conservation of gene expression allowed the separation of genes related to general sexual processes from ones restricted to CM species, providing functional hypotheses for genes potentially linked to sculpting CM fruiting bodies.

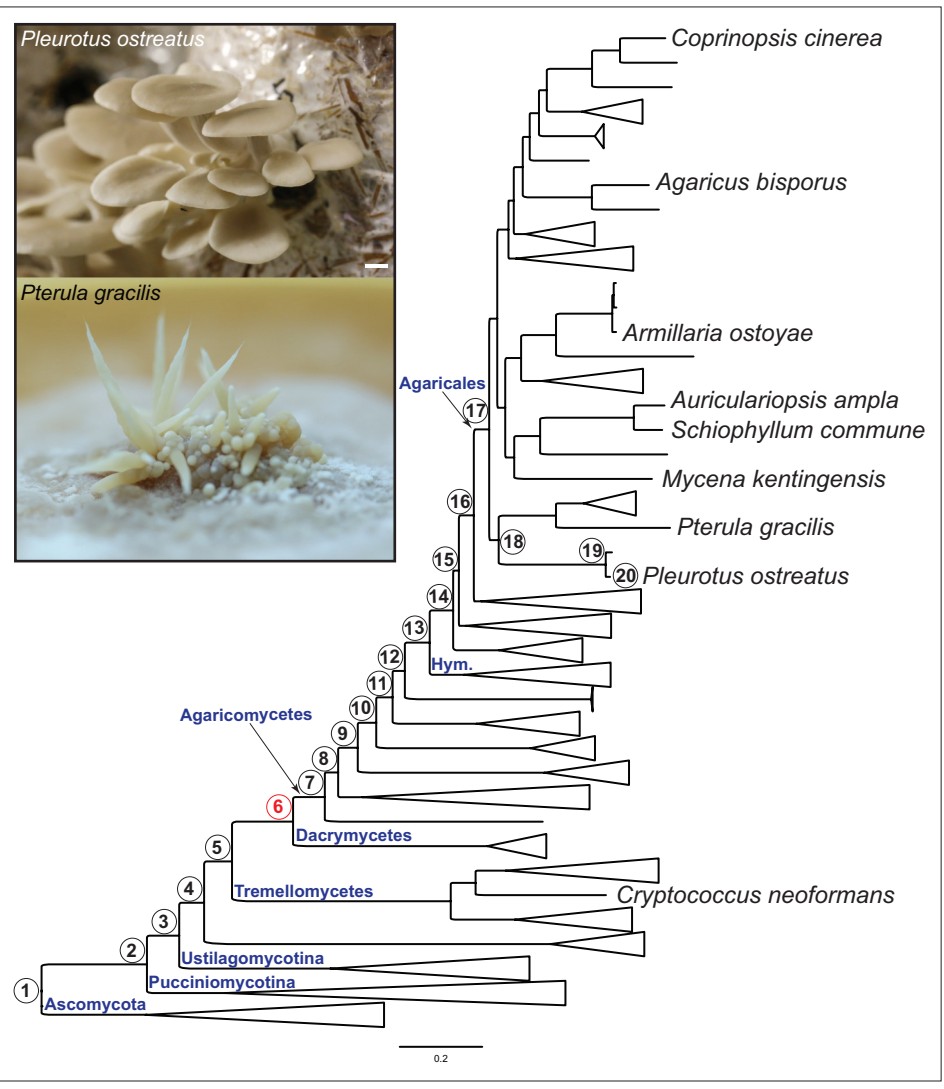

**Figure 1.** Pylostratigraphic gene ages and phylogenetic relationships among the nine species analyzed in this study. Numbers in circles next to nodes represent gene ages used in phylostratigraphic analyses of *Pleurotus ostreatus*. Nodes are numbered from 1 to 20 from the root of the tree to the tip harboring *P. ostreatus*. The first emergence of complex multicellularity in this lineage is shown with red, according to ***Merényi et al., 2020***. The scale bar represents 0.2 expected change per site. Fruiting bodies of *P. ostreatus* (upper) and *Pterula gracilis* (lower) are shown in the box. 'Hym,' Hymenochaetales.

The online version of this article includes the following figure supplement(s) for figure 1:

**Figure supplement 1.** Sampled developmental stages and tissue types during fruiting body formation of *Pleurotus ostreatus*.

**Figure supplement 2.** Sampled developmental stages during fruiting body formation of *Pterula gracilis*.

**Figure supplement 3.** Multidimensional scaling (MDS) plot based on the expression of genes in (**a**) *Pleurotus ostreatus* and (**b**) *Pterula gracilis*.

**Figure supplement 4.** The distribution of developmentally expressed genes in each species.

These data will help to understand both complex multicellular and simple sexual morphogenesis in Basidiomycete fungi.

## Results and discussion
### Overview of new RNA-seq data

We present highly resolved developmental transcriptome data for *P. ostreatus* (oyster mushroom), one of the three most widely cultured species worldwide (*Zhu et al., 2019*), as well as for *Pterula gracilis*, a closely related species with a simple fruiting body morphology (*Figure 1*). In *P. ostreatus,* we sampled six developmental stages and up to four tissue types within a stage, whereas in *Pt. gracilis*, tissues could not be separated; therefore, we sampled four developmental stages (*Figure 1—figure supplements 1 and 2*). Strand-specific RNA-seq yielded 15.9–34.0 million reads per sample (Dryad: Table D1). Multidimensional scaling of the normalized transcriptome data accurately identified sample groups with biological replicates being tightly positioned together (*Figure 1—figure supplement 3*). Fruiting body samples were grouped in two main groups, the early (primordia and young fruiting bodies) and mature fruiting bodies, irrespective of the tissue types. For uniformity in downstream analyses, we reanalyzed data from former studies (*Almási et al., 2019*; *Gehrmann et al., 2018*; *Ke et al., 2020*; *Krizsán et al., 2019*; *Liu et al., 2018*; *Sipos et al., 2017*), yielding data for eight species in the order Agaricales (*Figure 1*), which comprises a single origin of complex fruiting body morphologies (*Marisol et al., 2020*; *Varga et al., 2019*). *P. ostreatus* and *Pt. gracilis* had 4294 and 474 developmentally expressed genes (≥4 fold change [FC]), respectively. *Pleurotus* has a similar number of developmentally expressed genes to those reported earlier for other mushroom-forming fungi, while *Pt. gracilis* has fewer, possibly due to its simple morphology (*Almási et al., 2019*; *Krizsán et al., 2019*; *Sipos et al., 2017*; *Figure 1—figure supplement 4*). To validate the relevance of developmentally expressed genes, we collected experimentally validated, fruiting-related genes from *P. ostreatus* and the model species *Coprinopsis cinerea.* For these genes, 92.3% of the *P. ostreatus* orthologs showed developmental expression (at FC > 2) in our dataset (*Supplementary file 1*), indicating that our approach captures CM-related genes with high sensitivity.

### Developmentally expressed genes, natural antisense transcripts, and gene age distribution

Developmentally expressed genes displayed limited physical clustering in the genomes (Appendix 1, Dryad: Table D2), which is different from some key genes involved in animal and plant pattern formation (*Meyerowitz, 2002*). Notably, some of the developmental gene 'hotspots' overlapped with putative natural product biosynthetic gene clusters, a well-known group of clustered genes in fungal genomes (*Keller, 2019*).

In addition to protein coding genes, strand-specific RNA-seq data allowed us to annotate NATs in the transcriptomes of *P. ostreatus* and *Pt. gracilis* (Appendix 2). NATs were abundant in both species (2043 and 763 in *P. ostreatus* and *Pt. gracilis*, respectively), consistent with a previous report (*Ohm et al., 2010*) and showed dynamic developmental expression. However, they showed very little conservation across species, which potentially stems from fast evolution and/or recent origins. It has been proposed that NATs can arise from random promoters as transcriptional noise (*Lloréns-Rico et al., 2016*), a possibility that may be true for several or most, but probably not for all, NATs in *P. ostreatus* and *Pt. gracilis.* The cryptic nature of NATs hardly allows functional inferences to be made (e.g., based on correlated expression with sense genes, see Appendix 2), yet their recent origins and expression patterns suggest that they may be a source of developmental innovation at small timescales (Appendix 2).

To understand the composition of developmental transcriptomes, we sorted all protein coding genes (including developmentally expressed ones) using a phylostratigraphic approach, in which gene ages are assigned based on the set of species that possess clear orthologs (see Materials and methods). We found that developmental transcriptomes showed a clear gene age patterns: they are dominated by old and young genes in all species, creating 'U'-shaped distributions (*Figure 2*). This shape simply mirrors the genome-wide gene age distribution, indicating that the genomes of the examined species are dominated by conserved and young genes. If we statistically corrected for these U-shaped gene age distributions, we found that genes displayed an enrichment of developmental

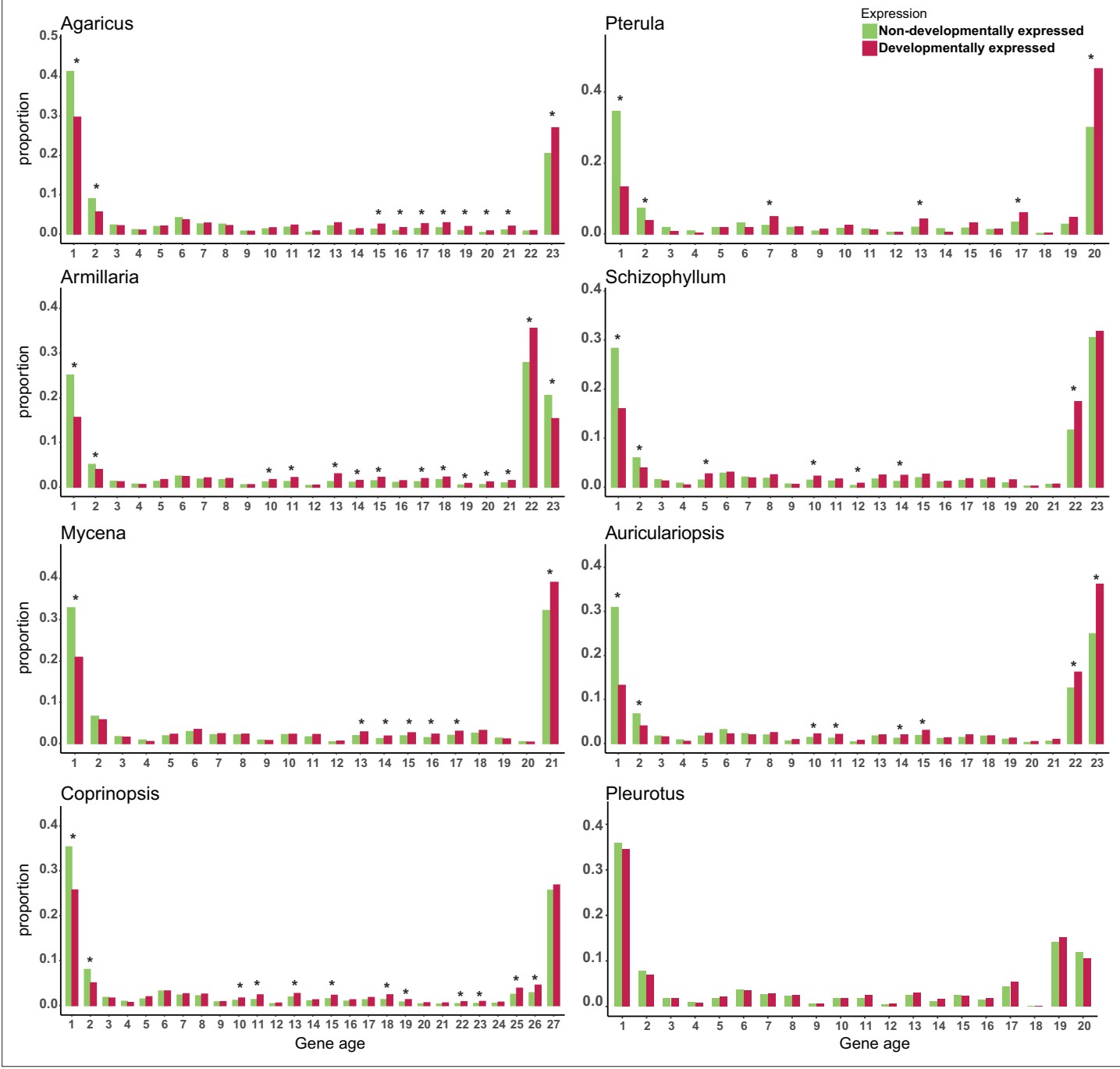

**Figure 2.** Proportion of developmentally expressed (DR >4 fold change [FC]) genes in different gene ages. * represents significant differences p-value<0.05 (Fisher's exact test with Benjamini–Hochberg correction). Gene age was calculated based on orthogroup membership (the presence of clear orthologs across species). For each species, nodes along the node path were numbered in ascending order on the species tree from root to tip starting with the value 1 (see *Figure 1* for an example); node numbers were then used as the gene age values.

expression (at FC > 4) among young genes in most species (Fisher's exact test, FDR-corrected p<0.05, *Figure 2*), indicating that young genes have a disproportionately high share among developmentally expressed ones in fruiting bodies. This could be either because these young genes are needed for sculpting fruiting body morphologies or because in young genes neutrally arising expression variation (i.e., transcriptional noise) is better tolerated than in conserved ones and leads to patterns we recognize as developmental expression.

## Developmental hourglass

To examine how young genes contribute to the CM transcriptome, we calculated transcriptome age indices (TAIs) for each developmental stage in each species. TAI calculations weigh phylostratigraphic patterns by expression level, thus providing a weighted view of the contribution of young and old genes to the transcriptome. This way, TAIs link gene ages to the developmental hourglass concept, which has been proposed to explain the incorporation of genetic novelty into the developmental programs of CM eukaryotes (*Domazet-Lošo and Tautz, 2010*; *Drost et al., 2017*), including fungi (*Cheng et al., 2015*). The hourglass concept posits that evolutionarily older genes are expressed at mid-development (*Domazet-Lošo and Tautz, 2010*) while the alternative 'early conservation' model implies that old genes are expressed early in development (*Piasecka et al., 2013*). Fungi do not display developmental transitions (e.g., phylotypic stage, mid-developmental transition) similar to those of metazoans, but they have a complex developmental program, and it has been proposed that the hourglass phenomenon would arise in any species with a sufficiently complex development (*Domazet-Lošo et al., 2017*). In fungi, the emergence of fruiting body primordia on vegetative mycelia comprises the largest developmental transition; dimensions change from fractal-like in mycelia to 3D in fruiting bodies, which necessitates turning on several traits for CM (*Nagy et al., 2018*). Accordingly, the largest transcriptomic reprogramming (e.g., in terms of differentially or developmentally

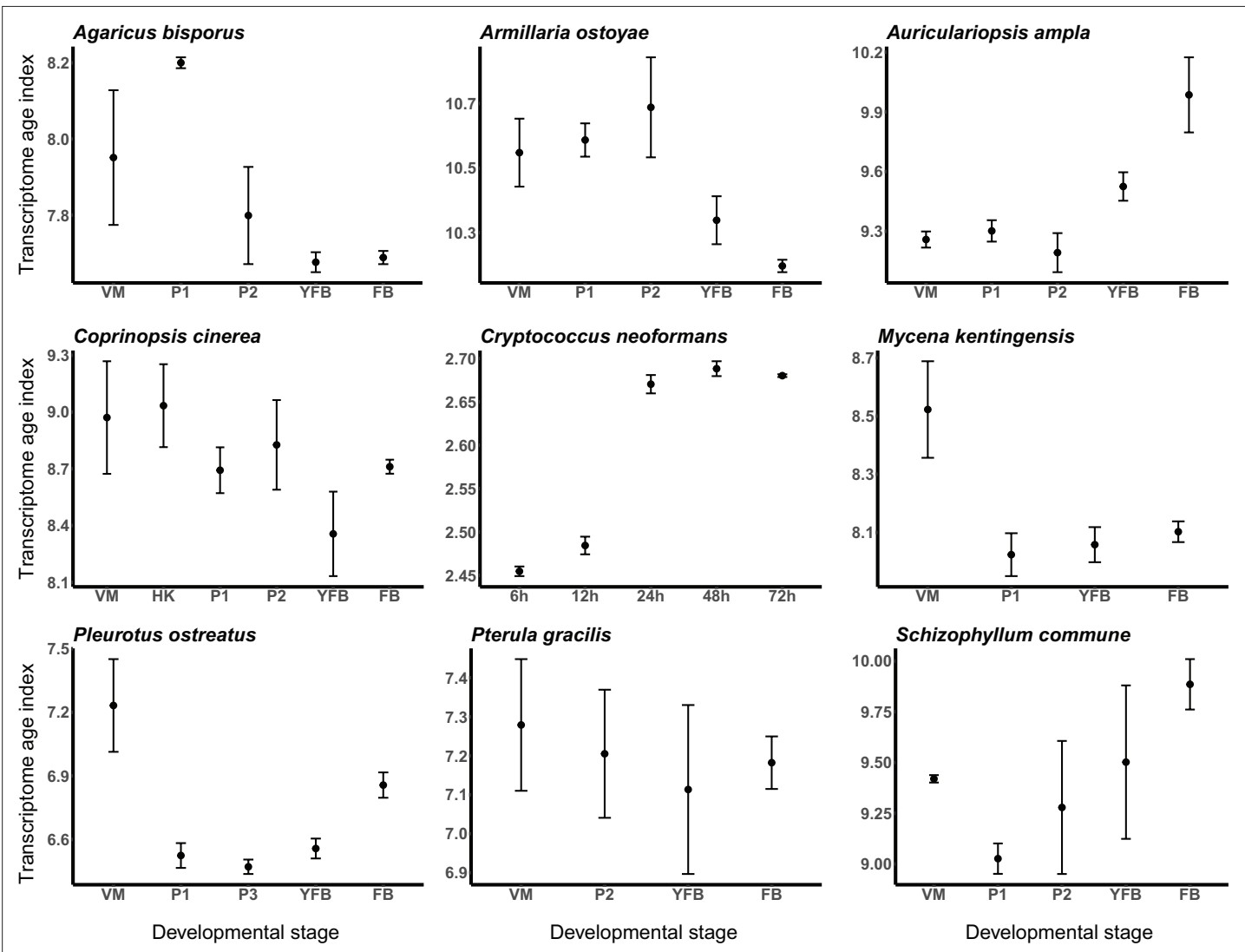

**Figure 3.** Transcriptome conservation in the nine species based on transcriptome age index (TAI). VM: vegetative mycelium; P1: stage 1 primordium; P3: stage 3 primordium; YFB: young fruiting body; FB: fruiting body.

expressed genes) was reported for this transition (**Krizsán et al., 2019**; **Sipos et al., 2017**; **Muraguchi et al., 2015**).

To test if a developmental hourglass can be found in fungi, we analyzed the transcriptomes of the nine species based on TAIs (see Materials and methods). For the examined species, we did not obtain uniform signal for either the hourglass or the early-conservation model (**Figure 3**). In other words, based on the TAI profiles, it appears that young genes (which drive TAI to higher values) do not have a uniform contribution to fruiting body transcriptomes across the examined species. In *Auriculariopsis ampla*, *C. neoformans,* and *Schizophyllum commune,* TAI values were lowest early in development, compatible with the early conservation model, whereas in *Armillaria ostoyae*, *Agaricus bisporus*, *C. cinerea*, *Mycena kentingensis,* and *Pt. gracilis* the opposite pattern (TAIs highest in early development) was observed. Typical hourglass-like patterns were seen only in *P. ostreatus*, which seems to be an exception among the examined species (**Figure 3**).

Overall, we interpret these results as evidence for neither the hourglass nor the early-conservation models being applicable to mushroom development. Complex multicellular fruiting bodies of fungi, to our best knowledge, do not undergo the key developmental transitions (e.g., phylotypic stage, mid-developmental transition) like animal embryos do (**Virágh et al., 2021**). Rather, they follow unique developmental programs, which are discussed in the second half of the article.

## Allele-specific expression, but not RNA editing, is abundant in fruiting bodies of *P. ostreatus*

ASE and RNA editing are two processes that can shape the transcriptome by altering abundances and sequences of transcripts, respectively. Both have recently been reported in CM fungi (**Gehrmann et al., 2018**; **Zhu et al., 2014**), but how widespread they are and how they contribute to fruiting body development is poorly known. We chose *P. ostreatus* to analyze the contributions of ASE and RNA editing because both parental genomes have been sequenced (**Alfaro et al., 2016**; **Riley et al., 2014**) and sufficiently differ from each other to classify single-nucleotide variants either as ASE (variants differing from one parental genome) or RNA editing (variants differing from both parental genomes).

Overall, 2,244,348 variants served as input to the ASE analysis and were used to decide which haploid nucleus the reads originated from (Dryad: Table D3). We inferred that 31.2% and 32.2% of the reads derive from one (PC15) and the other (PC9) haploid parental nucleus, respectively, while 36.5% of reads were not assigned to either parental genome (Dryad: Table D3). This allowed us to characterize 10,419 PC15 genes (84.5% of all genes and 96.8% of expressed genes) for ASE. Similar to gene expression, ASE levels showed clear stage- and tissue-specific patterns (**Figure 4—figure supplement 1**).

At the scale of the entire genome or scaffolds, the two parental genomes expressed almost equally (**Figure 4—figure supplement 2**), whereas at the gene level 7793 (74,8%) of the 10,419 expressed genes were assigned as equally expressed genes (EE genes) in all stages and tissue types and 2626 genes (25.2%) were biased toward the same nucleus in all biological replicates of at least one stage or tissue (referred to as ASE genes, **Figure 4**). Of these, 1560 genes showed 2 < FC < 4 fold expression imbalance (hereafter referred to as S2 genes; 15%) and 1066 showed over fourfold difference (S4 genes; 10.2%) between the two nuclei in at least one stage (averaged across replicates, Dryad: Table D4). In comparison, in *A. bisporus* ASE was reported for 411 genes (~4% of the genome), perhaps due to fewer SNPs between parental nuclei (**Gehrmann et al., 2018**).

Enrichment analysis based on InterPro (IPR) domains and Gene Ontology (GO) terms of ASE genes highlighted a significant overrepresentation of 83 IPR and 45 GO terms, respectively (**Supplementary file 2a–d**), several of which are associated with genes known to be involved in fruiting body formation (**Krizsán et al., 2019**), such as hydrophobins, glycoside hydrolase families, aquaporins, and fungal-type protein kinases (**Supplementary file 2a–d**). For example, we detected ASE in hydrophobin genes (**Figure 5a**), which are one of the most studied fruiting body-related gene families (**Bayry et al., 2012**). Both the fungal-type cell wall GO term (GO:0009277) and the hydrophobin-related (IPR001338) terms were significantly overrepresented among genes with ASE. Of the 27 hydrophobins of *P. ostreatus,* 21 showed developmental regulation (FC > 2), of which 14 showed ASE (FC > 2). Mycotoxin biosynthetic process (GO:0043386) was also enriched in both the GO and IPR analyses (**Figure 5b**). *Pleurotus* has 16 genes in the UstYa-like mycotoxin biosynthesis protein family (probably involved in dikaritin production; **Vogt and Künzler, 2019**), of which 6 were developmentally expressed (FC > 4), and all of

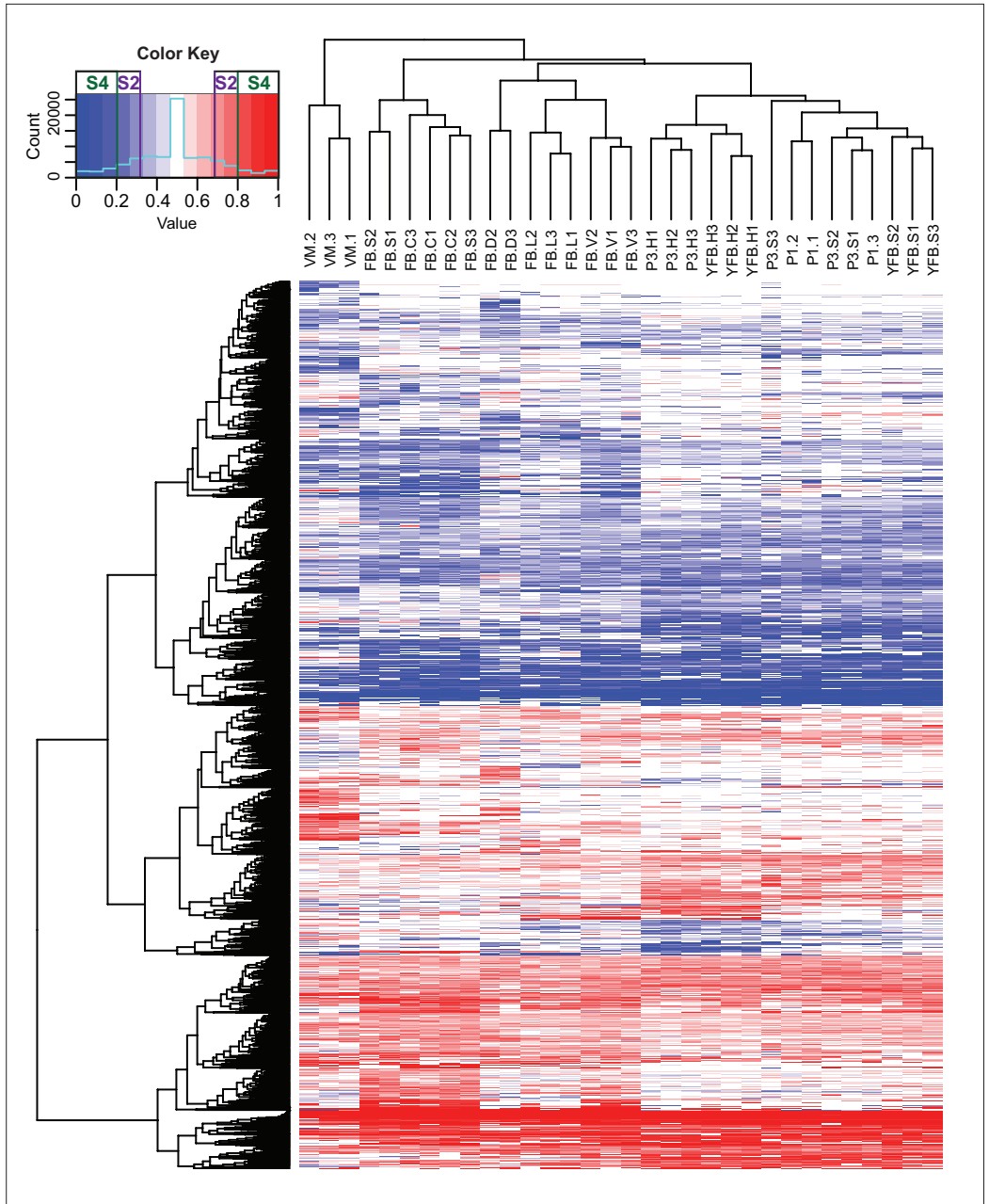

**Figure 4.** Contribution of two haploid nuclei of *P. ostreatus* to total gene expression. Expression of PC15 relative to the sum of PC15 and PC9 (AS ratio) was visualized in a heatmap for genes that showed at least twofold allele-specific expression (ASE) in at least one stage. Thresholds that we used to define S2 and S4 gene sets are marked in the color key. VM: vegetative mycelium; P1: stage 1 primordium; P3: stage 3 primordium; YFB: young fruiting body; FB: fruiting body; H: cap (entire); C: cap trama; L: lamellae; S: stipe; V: cuticle; D: dedifferentiated tissue of cap.

The online version of this article includes the following figure supplement(s) for figure 4:

**Figure supplement 1.** Principal component analysis based on AS ratio.

**Figure supplement 2.** Allele-specific expression was not biased toward one nucleus or chromosome(s).

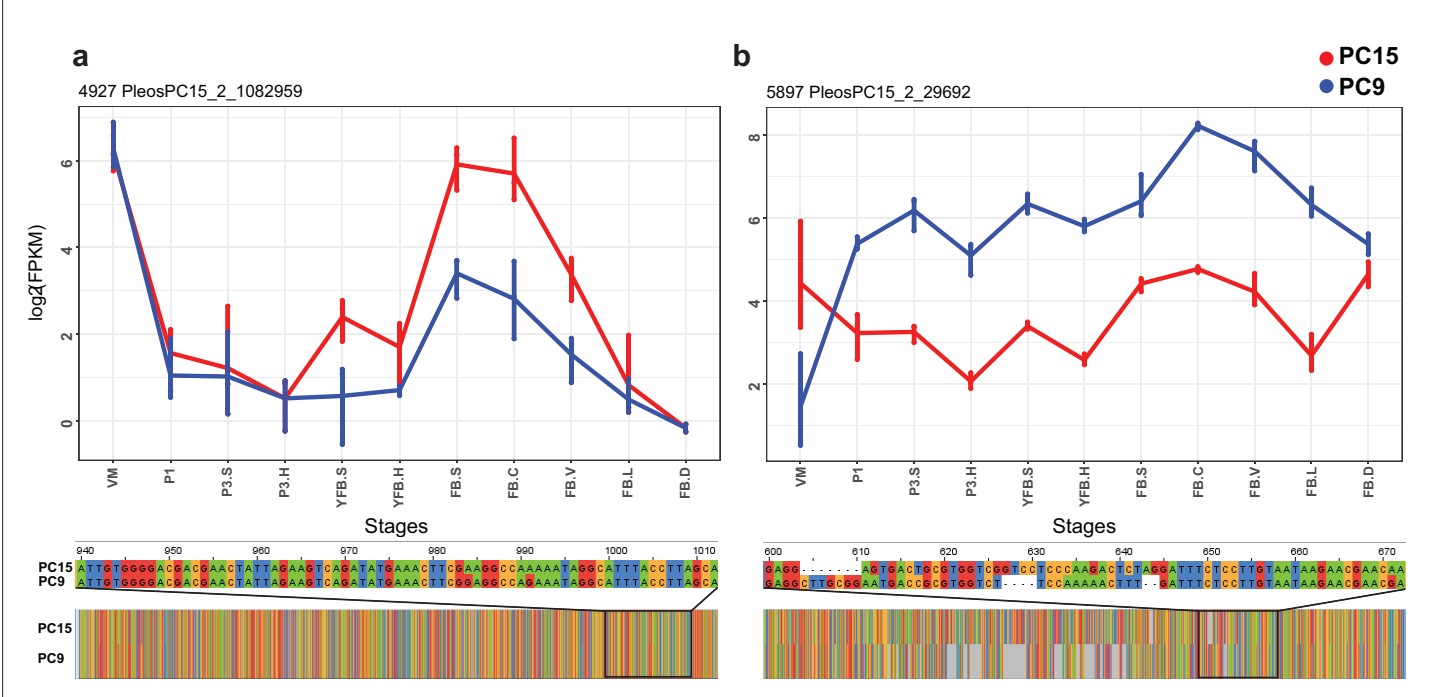

**Figure 5.** Examples of allele-specific expression (ASE) during fruiting body formation of *Pleurotus ostreatus*. Expression level (log₂ transformed fragments per kilobase of transcript per million mapped reads [FPKM]) from the two nuclei are colored with blue (PC9) and red (PC15). *P. ostreatus* gene- and protein IDs (PleosPC15_2) are displayed in each plot as a title. (**a**) Hydrophobin; (**b**) UstYa-like mycotoxin biosynthesis protein genes. Differences in the upstream gene regions are shown under the plots. VM: vegetative mycelium; P1: stage 1 primordium; P3: stage 3 primordium; YFB: young fruiting body, FB: fruiting body; H: cap (entire); C: cap trama; L: lamellae; S: stipe; V: cuticle; D dedifferentiated tissue of cap.

these showed ASE, while only 1 of the remaining 10 genes showed ASE. These examples highlighted how ASE can generate expression variance in developmentally expressed genes and thus potentially influence fruiting-related gene expression.

Adenosine-to-inosine (A-to-I) RNA editing is another source of single-nucleotide variants in the transcriptome that was recently described in CM fungi (*Zhu et al., 2014*; *Liu et al., 2017*; *Bian et al., 2019*; *Teichert, 2020*). In contrast to ASE, however, our analyses did not yield considerable signal for A-to-I editing in *P. ostreatus* (Appendix 3, Dryad: Table D5). In fact, most candidate sites turned out to be likely sequencing errors or hallmarks of polyadenylation sites (see Appendix 3 for details), indicating that RNA editing is probably not associated with fruiting body development in this species.

## Allele-specific expression is enriched in young genes

We next asked what mechanisms could give rise to ASE. *Gehrmann et al., 2018* found that DNA methylation can explain at most 10% of ASE, which is consistent with the negligible role of gene body methylation in fungi (*Montanini et al., 2014*), suggesting other mechanisms. Following reports of divergent *cis*-regulatory alleles causing allelic gene expression imbalance (*Gaur et al., 2013*; *Chen et al., 2016*; *Cowles et al., 2002*; *McManus et al., 2010*; *Wang et al., 2017*), we hypothesized that ASE may arise from *cis*-regulatory divergence between nuclei of *P. ostreatus*. The dikaryotic stage of fungi, in which two haploid nuclei coexist in the same cellular compartment, represents a compatible environment for ASE to arise. Indeed, upstream 1 kb regions, which presumably contain *cis*-regulatory elements, of S2 and S4 genes are significantly more different (Kruskal–Wallis with Nemenyi post hoc test $p<2e^{-16}$) between the two parents than upstream regions of EE genes (*Figure 6a*). This raises the possibility that divergent *cis*-regulatory elements in the same *trans*-regulatory cellular environment cause differential binding of transcription factors, resulting in biased transcript accumulation from the two nuclei. Amino acid sequences of S4 and S2 genes are also significantly more different between the two parents (Kruskal–Wallis test with Nemenyi post hoc test $p<2.2e^{-16}$; *Figure 6—figure supplement 1a*) than those of EE genes. Together, these observations indicate that ASE in *P. ostreatus* may

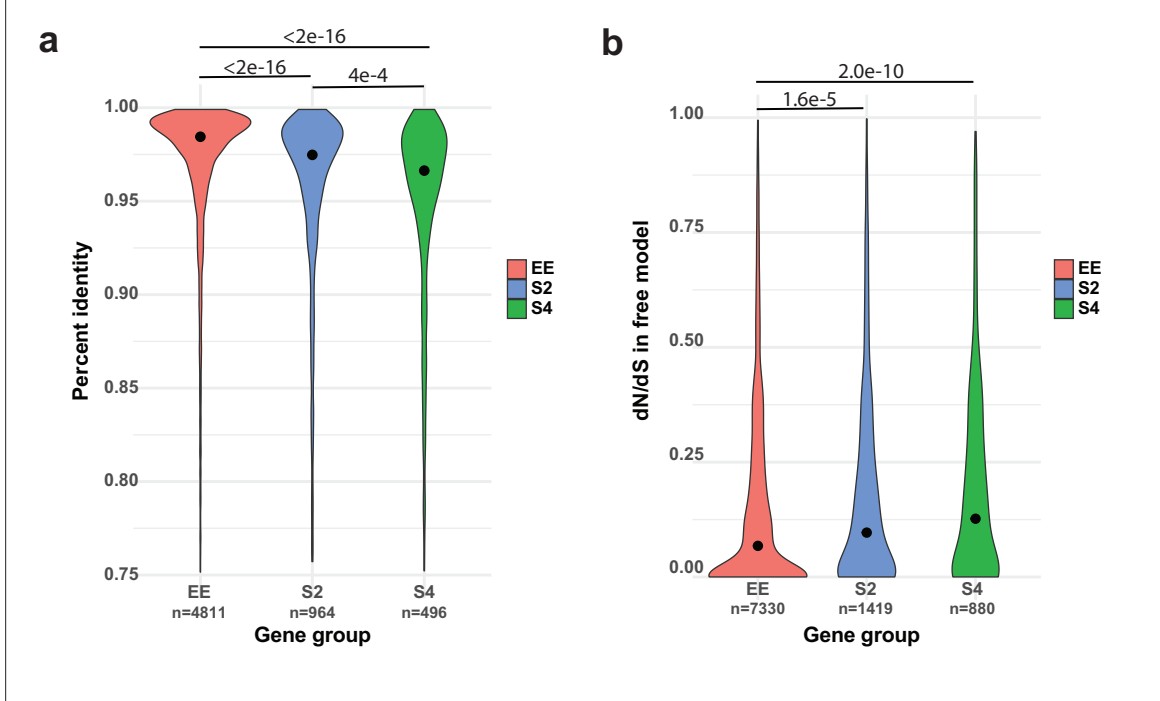

**Figure 6.** Allele-specific expression (ASE) may arise from *cis*-regulatory divergence. (**a**) Percent sequence identity between the 1 kb upstream regions of PC15 and PC9 genes. (**b**) dN/dS distribution for ASE (ASE with twofold change [S2] and ASE with fourfold change [S4]) and equally expressed (EE) genes under the free model in CodeML.

The online version of this article includes the following figure supplement(s) for figure 6:

**Figure supplement 1.** Comparison of genes with allele-specific expression (ASE) or equally expressed (EE).

arise from the divergence of *cis*-regulatory alleles, possibly in fast-evolving genes. Analysis of selection strength based on dN/dS ratios indicated higher dN/dS and thus weaker selection among ASE than among EE genes (Kruskal–Wallis test with Nemenyi post hoc test p=2.0e$^{-10}$ and 1.6e$^{-5}$; *Figure 6b*, *Figure 6—figure supplement 1b*), suggesting that ASE is enriched in genes that are released from selection constraints.

A well-known group of genes under relaxed selection are evolutionarily young genes; that is, those that duplicated or arose via de novo gene birth recently. Therefore, we tested whether ASE is correlated with relative gene age in our dataset. ASE genes were strongly and significantly overrepresented in the youngest gene ages (Fisher's exact test p=1.1e$^{-12}$–2.4e$^{-68}$), with a clear trend (*Figure 7/a*, Mann–Kendall test p=2.5e$^{-6}$) of increasing ASE incidence toward young genes. At the same time, ASE is significantly underrepresented in the oldest age categories (gene age 1–4: p=1.2e$^{-2}$–7.8e$^{-103}$, Fisher's exact test, *Figure 7a*). These observations are consistent with young genes tolerating allelic expression imbalance better than conserved ones, possibly due to relaxed constraint (*Dong et al., 2011*; *Gu et al., 2005*; *Kondrashov et al., 2002*).

If genes under weak selection can tolerate expression variation, and developmental expression is considered an adaptively or neutrally arising expression variation, then ASE genes and developmentally expressed genes should overlap to some extent. Indeed, half of the ASE genes (S4: 52.7%; and S2: 49.1%) were also developmentally expressed (FC > 4), significantly more than in EE genes (31.8%, Fisher's exact test p=8.2e$^{-58}$). We observed that as we move toward younger genes the proportion of developmentally expressed ASE genes increases compared to non-ASE genes (*Figure 7b*, *Figure 7—figure supplement 1*). The strongest overrepresentation of ASE genes can be observed among developmentally expressed genes that arose in the genus *Pleurotus* (gene age 19–20, p$_{S4/EE}$=3.0e$^{-20}$-9.52e$^{-14}$, Fisher's exact test).

Taken together, the above observations allow us to speculate that ASE is, to a large extent, likely arising as a neutral phenomenon. Accordingly, we see two implications on the interpretation of fruiting body transcriptome data. First, it is possible that some of the developmental variation generated by

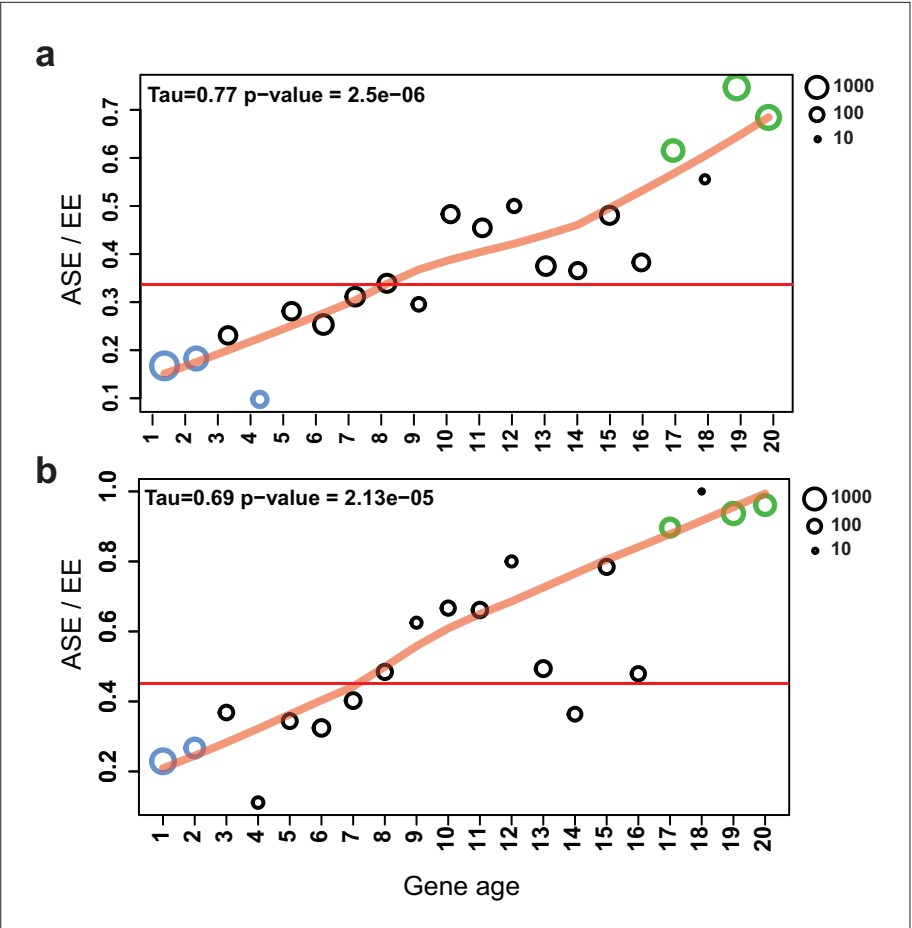

**Figure 7.** Allele-specific expression (ASE) genes are enriched among young genes. The proportion of ASE shows a significant tendency toward higher values (Mann–Kendall statistics) across gene ages in the case of (**a**) all genes and (**b**) developmentally expressed genes (fold change [FC] > 4). The horizontal red line represents the ratio of all ASE/equally expressed (EE) genes, while significant differences from the background (Fisher's exact test p-value<0.001) are shown with green (overrepresentation) and blue (underrepresentation). The size of circles represents the number of proteins ($\log_{10}$ transformed).

The online version of this article includes the following figure supplement(s) for figure 7:

**Figure supplement 1.** Distribution of genes across gene ages (1 representing the oldest and 20 the youngest), broken down by allele-specific expression (ASE), developmental regulation, and duplication in *Pleurotus ostreatus*.

---

ASE can prove adaptive at small evolutionary scales, which may manifest as between-strain differences within a species. Alternatively, ASE may be tolerated in genes with limited or species-specific functions, in which case it may have no or weak phenotypic impact on CM fruiting bodies. The overlap of ASE and developmentally expressed genes further suggests that developmentally expression in young genes can partially be neutrally arising expression variance as well.

## Comparative transcriptomics defines core developmentally expressed genes in the Basidiomycota

We have shown that the high number of young developmentally expressed genes could be either the result of neutral transcriptional variation and/or might be responsible for species specific functions. Therefore, we hereafter focus on conserved developmentally expressed genes to characterize core functions and gene families associated with the development of CM fruiting bodies.

Fruiting bodies encompass multiple processes, including sexual spore formation, defense, and tissue differentiation, among others, but only some of these are relevant from the perspective of the origin of CM. To identify core fruiting-related genes that participate in the sculpting of fruiting bodies

in Agaricomycetes in general, young and/or species-specific genes and genes with species-specific developmental expression need to be eliminated from the transcriptomes. To remove young genes, we first looked for sets of 1-to-1 orthologs across the examined species, hereafter called 'orthogroups' (one gene per species), which show developmentally dynamic expression in most species (FC > 2/4, see Materials and methods). This yielded 1781 orthogroups, considered hereafter as conserved developmental orthogroups.

To distinguish genes related to basic sexual processes (sporulation, meiosis) from those restricted to CM fruiting bodies, we reanalyzed transcriptome data for sexual sporulation and basidium development of *C. neoformans* (*Liu et al., 2018*). This species is closely related to the Agaricomycetes and has a simple, non-CM development, so we used it here as a minimal model of sexual development (*Figure 8a*). Of the 1781 conserved orthogroups, 913 and 868 were developmentally expressed both in *C. neoformans* and CM species and only in CM species (*Figure 8b*, *Supplementary file 3a–c*), and are referred to as shared and CM-specific orthogroups, respectively. Of the 868 CM-specific orthogroups, 754 were completely missing in *C. neoformans,* whereas 114 were present but not developmentally expressed (*Figure 8b*, *Supplementary file 3c*). The 754 orthogroups might be missing from *C. neoformans* because they evolved later (in Agaricomycetes) or because they were lost during the reductive evolution of this species. Shared orthogroups included highly conserved gene functions, such as mitosis/meiosis, general transcription factors, or ribosomal proteins, whereas CM-specific orthogroups contained more genes encoding sequence-specific transcription factors, cell wall remodeling, oxylipin biosynthesis, protein ubiquitination (F-box, BTB/POZ, and RING zinc finger domain proteins), as well as functionally unclassified proteins (*Figure 8c, Figure 8—figure supplement 1*, *Supplementary file 3b*).

Cell division-related (DNA replication, meiosis, mitosis, DNA repair, etc.) and ribosomal protein encoding genes comprised the most frequent annotations in shared orthogroups (*Figure 8d*, *Figure 8—figure supplement 1*). Meiosis happens in basidia in both *C. neoformans* and fruiting body forming fungi and associated genes showed clear peaks in their expression (*Figure 8—figure supplement 2*). *C. neoformans* showed a single peak in meiotic/mitotic gene expression, whereas CM fungi showed two peaks, one corresponding to meiosis in gills and another to intense cell division (mitosis) in primordia. Ribosomal protein gene expression, as a proxy for the activity of protein synthesis, has been widely associated with cell growth and proliferation (*Jorgensen et al., 2002*; *Kraakman et al., 1993*). Ribosomal proteins showed an early peak in all species, while in CM species a second peak was also observed, coincident with meiosis and spore production in gills, suggesting increased protein synthesis (*Figure 8—figure supplement 3*). We infer that in CM species the first ribosomal gene expression peak corresponds to an early, proliferative phase of development followed by the transition to growth by cell expansion (*Krizsán et al., 2019*), which gives the final shape and size of fruiting bodies before spore release.

Several cell surface proteins (fasciclins, ricin-B lectins, and the PriA family) and putative cell wall remodeling enzymes (e.g., chitin- and glucan- active glycoside hydrolases, expansins, CE4 chitoligosaccharide deacetylases, laccases) previously linked to fruiting body morphogenesis (*Pezzella et al., 2013*; *Xie et al., 2018*) were shared between *C. neoformans* and fruiting body forming species (*Supplementary file 3d*), suggesting that these families are important for sexual morphogenesis in general, not restricted to fruiting bodies, as thought previously. Cell wall remodeling enzymes have been hypothesized to produce fruiting body-specific cell wall architectures (*Buser et al., 2010*; *Krizsán et al., 2019*; *Liu et al., 2021*; *Ohga et al., 2000*); the upregulation of these in *C. neoformans* suggests a role during non-CM sexual processes as well, possibly in generating aerial hypha- or basidium-specific cell walls. Most genes related to glycogen metabolism also showed shared expression (*Supplementary file 3f*). Glycogen has been known as a storage material in fruiting bodies, but our observations indicate that it may serve that role in *C. neoformans* too and possibly as an energy source for sexual development or as storage carbohydrate in spores, in general. Notable transcription factors in shared orthogroups included the light sensing white collar complex member WC-1, orthologs of *Saccharomyces cerevisiae* sexual reproduction-related Ste12, a Basidiomycota-specific velvet factor, as well as orthologs of the carbon catabolite repressor CreA of *Aspergillus nidulans*.

In comparison to shared orthogroups, CM-specific orthogroups contained more transcription factors, genes related to cell wall biosynthesis/modification and defense (*Figure 8—figure supplement 1*, *Supplementary file 3d–g*). 33 CM-specific orthogroups of transcription factors were detected,

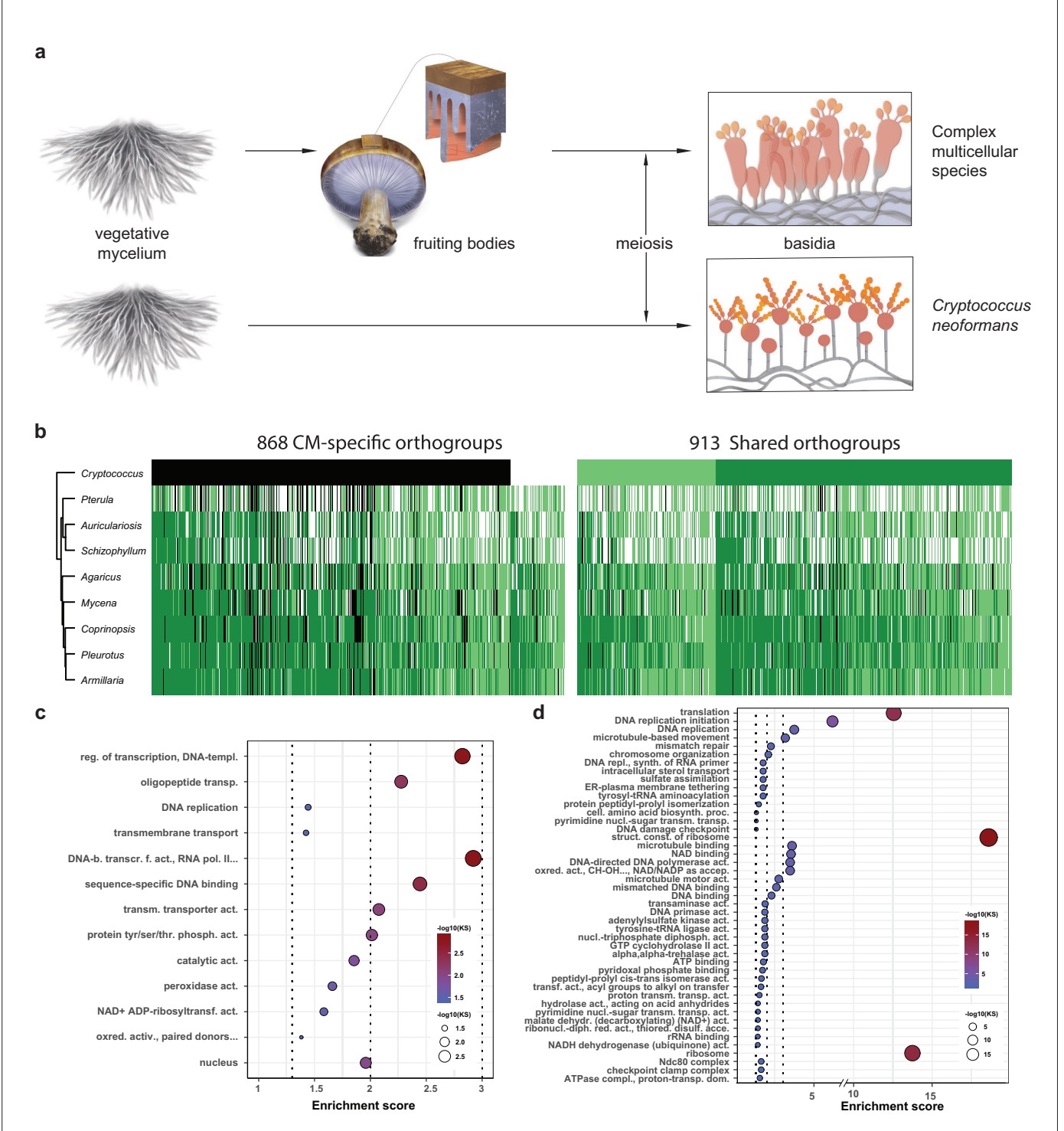

**Figure 8.** Conserved developmental expression in complex multicellularity (CM) fungi. (**a**) Schematic representation of complex multicellular and simple development and the use of *C. neoformans* as a minimal model of sexual development in the Basidiomycota. (**b**) Distribution of genes and their developmental expression across the nine species. Dark and light green refers to genes with developmental regulation at fold change > 4 and 2, respectively, whereas white and black denote nondevelopmentally expressed and missing genes, respectively. Dendrogram was inferred in a hierarchical clustering based on expression categories. Gene Ontology (GO) enrichment for (**c**) CM-specific and (**d**) shared orthogroups. KS means the p-value of Kolmogorov–Smirnov test implemented in the R package 'topGO.' On panels (**c**) and (**d**), cutoff lines (dashed line) are drawn at enrichment scores corresponding to p=0.05, p=0.01, and p=0.001 (from left to right). GO terms are ordered by Kolmogorov–Smirnov p-values. See also ***Supplementary file 5a and b*** for GO enrichment details.

*Figure 8 continued on next page*

*Figure 8 continued*

The online version of this article includes the following figure supplement(s) for figure 8:

**Figure supplement 1.** Functional categories across the conserved developmental orthogroups.

**Figure supplement 2.** Expression of meiotic genes in the nine species.

**Figure supplement 3.** Expression of ribosomal proteins in the nine species.

including those containing the hom1, fst3, and fst4 genes of *S. commune*, which were reported to influence the formation of fruiting bodies (*Ohm et al., 2011*; *Figure 8—figure supplement 1*). Based on our 109-species dataset, these three genes evolved after *C. neoformans* (Tremellomycetes) split off from CM fungi: Fst4 and Hom1 emerged in the most recent common ancestor (MRCA) of Agaricales and Gomphales (node 11 in *Figure 1*) while Fst3 appeared in the MRCA of Agaricales and Auriculariales (node 10 in *Figure 1*). Hydrophobins and cerato-platanins, as well as fatty acid desaturases and linoleate-diol synthases, were exclusively found in CM-specific orthogroups (*Figure 8—figure supplement 1*). Hydrophobins and cerato-platanins are cell surface proteins that confer hydrophobicity to hyphae and are completely missing from the genome of *C. neoformans*, probably as a consequence of the adaptation to a primarily yeast-like lifestyle (*Nagy et al., 2014*). Fatty acid desaturases and linoleate-diol synthases are putatively related to the biosynthesis of signaling-related oxylipins (*Orban et al., 2021*) and linoleic acid (a fruiting body-enriched membrane constituent; *Sakai and Kajiwara, 2003*; *Song et al., 2018*), respectively. We also detected a large number of conserved unannotated genes (172 orthogroups) among CM-specific orthogroups. Unannotated genes include, for example, *S. commune* Spc14 and Spc33, which were shown to participate in septal pore cap formation in Agaricomycetes (*van Peer et al., 2010*), and Cc.ctg1 of *C. cinerea*, which was suggested to be required for stipe elongation (*Nakazawa et al., 2008*). These genes are conserved across Agaricomycetes, but do not contain any known conserved protein motifs. Functional speculations are hardly possible for the vast majority of unannotated orthogroups, yet their propensity among CM-specific genes underscores the still cryptic nature of CM development in fungi. The complete list of shared and CM-specific orthogroups is given in *Supplementary file 3b and c*; however, their comprehensive discussion is beyond the scope of this article.

CM-specific orthogroups showed a phylostratigraphic enrichment in early mushroom-forming fungi (FDR < 0.01, Fisher's exact test, *Figure 9*). We detected a preponderance of CM-specific orthogroup origins from the MRCA of Dacrymycetes and Agaricomycetes (node six on *Figure 1*) to that of Hymenochaetales and Agaricales (node 13 on *Figure 1*). These may correspond to innovations related to CM fruiting bodies, which is consistent with the origins of jelly-like fruiting bodies in the Dacrymycetes + Agaricomycetes ancestor (*Virágh et al., 2021*). This observation complements our previous analysis (*Krizsán et al., 2019*) that could not resolve a clear signal of genetic innovation correlated with CM, possibly because of confounding effects of shared orthogroups.

## Conclusions

In this study, we analyzed developmental transcriptomes of complex multicellular fungi in the Agaricomycetes using a comparative dataset that included the first well-resolved developmental gene expression profiling data for *P. ostreatus* (oyster mushroom), the second most widely cultured mushroom species worldwide (*Grimm and Wösten, 2018*; *Royse et al., 2017*). We detected evidence for widespread developmental expression of genes, ASE imbalance between parental monokaryons, NATs, but not for RNA editing or the developmental hourglass. We found that the detected phenomena affect genes of various evolutionary ages and speeds to different extents. For example, developmental expression and ASE were most pronounced among evolutionarily young genes. On the other hand, NATs showed no conservation across species, suggesting that they evolve at a high rate. These observations allow us to speculate that the complex interplay of these processes in the transcriptome may provide multiple gears for transcriptome evolution that probably facilitates the incorporation of evolutionary innovations into fruiting body development of Agaricomycetes.

The availability of the genomes of both parental monokaryons (*Alfaro et al., 2016*; *Riley et al., 2014*), as well as new strand-specific RNA-seq data, allowed bioinformatic deconvolution of RNA editing, ASE, and antisense transcription in the *P. ostreatus* transcriptome. We found virtually no evidence for RNA editing, whereas ASE was abundant, which supports a previous report of ASE in

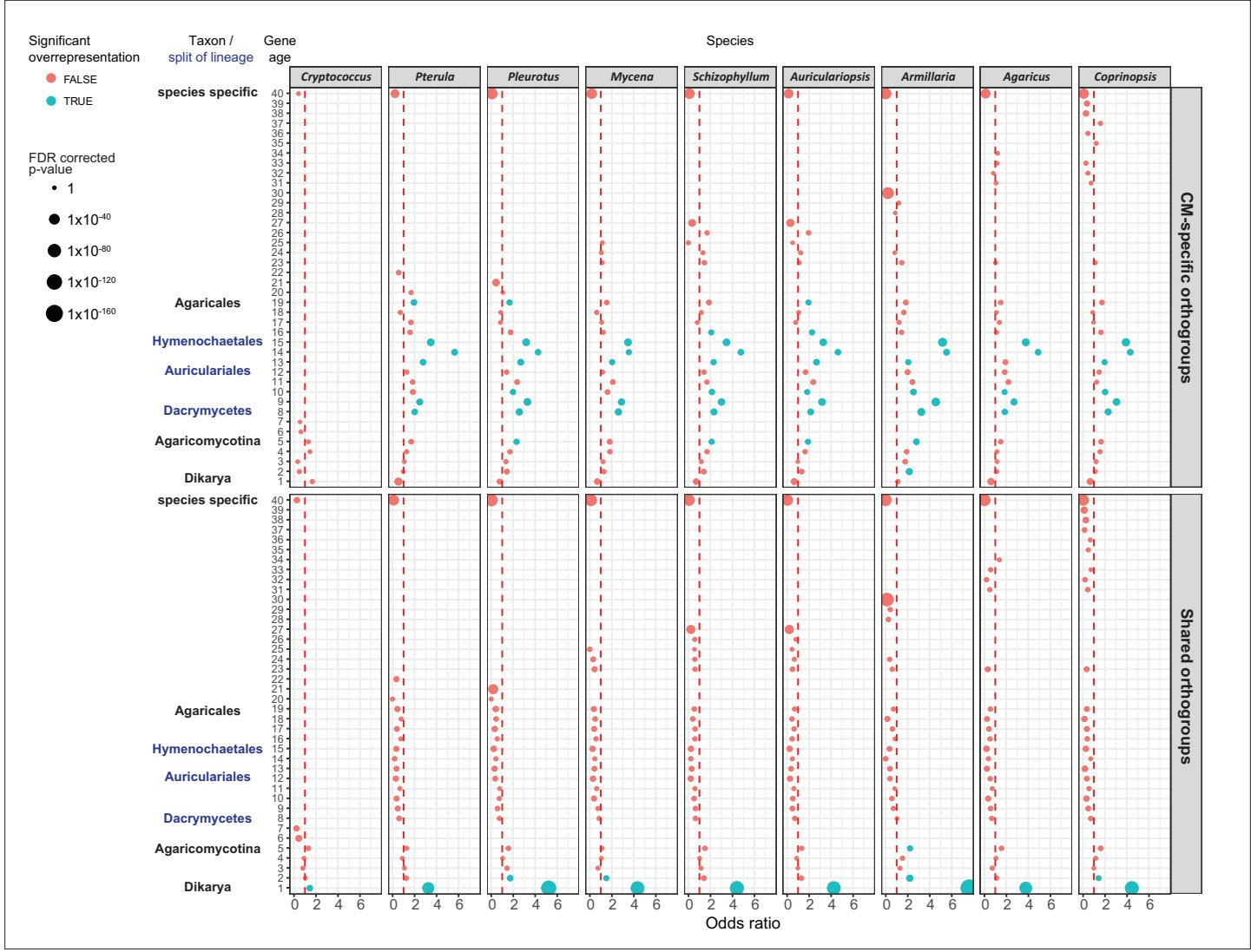

**Figure 9.** Gene age enrichment for shared and complex multicellularity (CM)-specific developmentally expressed orthogroups. CM-specific orthogroups are significantly enriched in the most recent common ancestors of lineages in which the first complex structures emerged (see also *Figure 1*). Y-axis represents relative gene age (for definition of gene ages, see Dryad: Figure D1). X-axis displays the odds ratio of the enrichment of developmentally expressed orthogroups relative to all orthogroups in a given age category based on Fisher's exact test. If the odds ratio exceeds 1 (red dotted line), developmentally expressed orthogroups are overrepresented in that gene age. Significant (FDR p<0.05) overrepresentation is indicated by blue. The size of circles corresponds to the FDR-corrected p-value of Fisher's exact test.

CM fungi (*Gehrmann et al., 2018*). RNA editing has been recently reported in the Agaricomycetes (*Zhu et al., 2014*); however, in contrast to the Ascomycota (*Lau et al., 2020*; *Liu et al., 2016*; *Teichert et al., 2017*), it displayed no clear-cut enrichment of A-to-I-compatible variants in three previous studies (*Bian et al., 2019*; *Teichert, 2020*; *Zhu et al., 2014*) or in this study. Rather, our final candidate RNA-editing sites merely alluded to potential polyA site- and/or read alignment inaccuracies, leading us to conclude that RNA editing is not abundant in *P. ostreatus*.

On the other hand, ASE was detected in thousands of genes in *P. ostreatus*. In a previous study on *A. bisporus,* ASE was interpreted as a regulated and adaptive mechanism that could, for example, aid the division of labor between nuclei in a dikaryotic hyphal cell (*Gehrmann et al., 2018*). We found that in *P. ostreatus* ASE is characteristic of young genes and likely arises from promoter divergence, which creates a cellular environment with divergent *cis*-regulatory alleles but identical *trans*-regulatory elements. At the same time, young genes are known to be under weaker evolutionary constraint than conserved ones, raising the possibility that ASE might arise neutrally in the transcriptome. This would be consistent with the neutral model of expression evolution (*Fay and Wittkopp, 2008*) and

nonadaptive explanations, such as leaky regulation or transcriptional noise (*Cheng et al., 2017*; *Khan et al., 2012*; *Shih and Fay, 2021*; *Wainer-Katsir and Linial, 2019*). Under this interpretation, ASE may be a tolerated, rather than an adaptive phenomenon in agaricomycete fungi. However, even if neutral at the level of the individual, ASE may generate useful gene expression variation that can serve as substrate for adaptive evolution (even for developmental functions), similar to how transcription from random promoters can facilitate de novo gene birth (*Van Oss and Carvunis, 2019*). Indeed, we detected ASE for several fast-evolving development-related genes, such as hydrophobins or the putatively defense-related dikaritin-synthesis family. ASE may have important implications in mushroom breeding, where intraspecific hybrids (e.g., *Gaitán-Hernández and Salmones, 2008*) harboring *cis*-regulatory alleles with various levels of divergence may show differences in industrially relevant traits (*Gehrmann et al., 2018*).

As in the case of ASE, young genes might display more expression variance and noise across development, whereas genes with conserved developmental expression more likely provide clues about key CM functions. Fruiting bodies integrate several ancient processes, such as mitosis/meiosis and sporulation, which are conserved across all organisms and fungi, respectively, but distinct from CM morphogenesis. These considerations led us to design analyses that remove both young and very ancient genes. These helped us distinguish conserved genes related to simple sexual development from those characteristic of only CM species. This may also help establishing a minimal model of sexual development (e.g., for pathogens like *C. neoformans*) in the Basidiomycota, which include several genes previously considered specifically expressed in fruiting bodies. Notable examples include fasciclins, which have been implicated in cell adhesion (*Nagy et al., 2018*), and the *PriA* family of secreted cell surface proteins (including *C. neoformans* cfl1 and dha1; *Gyawali et al., 2017*) with unknown function. On the other hand, this strategy yielded a focused set of 868 orthogroups that comprised genes developmentally expressed only in CM species not in *C. neoformans*. We speculate that these genes contain those related to CM morphogenesis, although this will need to be verified by functional studies in the future. Indeed, CM-specific orthogroups identified regulatory genes reported in mushrooms (e.g., hom1, fst3, fst4, wc1; *Hou et al., 2020*; *Kamada et al., 2010*; *Ohm et al., 2011*) but also novel ones, such as a velvet factor that is widely conserved in Agaricomycetes and showed stipe-specific expression in CM species. We anticipate that these orthogroups will comprise a valuable resource for functional studies of CM morphogenesis in fungi that, with continuous developments of genetic engineering methods, will make it possible to address the developmental roles of these genes at scale.

## Materials and methods

### Key resources table

| Reagent type (species) or resource | Designation | Source or reference | Identifiers | Additional information |
|---|---|---|---|---|
| Strain, strain background (*Pleurotus ostreatus*) | N001 | CETC | CECT-20600 | Wild-type dikaryotic strain |
| Strain, strain background (*Pterula gracilis*) | CBS 309.79 | CBS | CBS 309.79 | Wild-type dikaryotic strain |

### Growth condition, sampling, and RNA-sequencing

For fruiting the dikaryotic strain N001 (CECT-20600) of *P. ostreatus* (recently interpreted as *P. cf. floridanus*; *Li et al., 2019*), we first prepared spawn by inoculating sterilized rye and incubating for 10 days. Pasteurized straw-based commercial oyster compost (95 vol%) and the colonized spawn (5 vol%) were mixed gently, and 3 kg were filled into polyethylene bags. Bags were incubated in the dark at 27°C and 85–90% relative humidity for 17 days. Next, the bags were transferred to the growing room for fruiting at 18–19°C, relative humidity 80–85%, and 8/16 hr light/dark period (with approximately 1200 lux light intensity). We sampled vegetative mycelium (VM), six developmental stages, and five tissue types, each in three biological replicates as explained in *Figure 1—figure supplement 1*. VM was collected from the sawdust culture. We defined stage 1 primordia (P1) as the globose-triangular hyphae-covered structures without clearly recognizable differentiation; stage 2 (P2) primordia were defined as the first time point when caps were recognizable as pointed tips. Stage 3 primordia (P3) had a clearly differentiated and pigmented cap and an appearing fracture below the cap. The young fruiting body (YFB) stage was defined as the time point when the lamellae are clearly recognizable below the cap, and the diameter of the cap is less than 2 cm. Finally, in the

mature fruiting body stage (FB) lamellae are fully developed and meiosis/sporulation have started, and the cap expands (>5 cm). In the case of stage 1 and 2 primordia (P1 and P2), the whole tissue was collected containing both stipe and cap initials. In the stage 3 primordium (P3) and the YFB stages, stipes and caps were sampled separately. We divided mature fruiting bodies (FBs) into stipe (S), cap trama (C), cap cuticle (V), and gills (L). We defined cap (H) as the whole upper part of the fruiting body (in P3 and YFB) while cap trama (C) refers to just the inner part of cap without lamellae or cuticle (in FB) (see *Figure 1—figure supplement 1*). The last stage we sampled was the dedifferentiated cap trama (D), a dissection from inner cap tissue that was inoculated for 24 hr on a sterile PDA-agar plate until the emergence of new hyphae. Tissue from 3 to 8 individual fruiting bodies was pooled for each replicate of each sample type.

*Pt. gracilis* CBS 309.79 was inoculated onto Malt Extract Agar plates with cellophane and incubated at 25°C for 25–27 days. For fruiting, plates were moved to a growth chamber at 15°C under 10/14 hr light/dark period (light intensity: 11 µE m$^{-2}$ s$^{-1}$). VM samples were scraped off the cellophane after 3 days. We defined primordia (P) as small (<1 mm) globose structures, young fruiting bodies as ~5-mm-long awl-shaped structures, while structures longer than 10 mm were considered mature fruiting bodies (*Figure 1—figure supplement 2*).

Three biological replicates of each sample type were stored at –80°C until RNA extraction. Tissue samples were homogenized with micropestles using liquid N$_2$, and RNA was extracted by using the Quick-RNA Miniprep Kit (Zymo Research) according to the manufacturer's instructions. Strand-specific cDNA libraries were constructed from poly(A)-captured RNA using the Illumina TruSeq Stranded RNA-Seq library preparation kit and sequenced on the Illumina HiSeq 4000/x platform in PE 2 × 150 format with 40 million reads per sample at OmegaBioservices (USA).

## Bioinformatic analyses of RNA-seq data

New data for *P. ostreatus* and *Pt. gracilis* was reanalyzed together with previously published transcriptomes of seven Basidiomycota species (*Supplementary file 4*). To remove adaptors, ambiguous nucleotides, and any low-quality read end parts, reads were trimmed using bbduk.sh and overlapping read pairs were merged with bbmerge.sh tools (part of BBMap/BBTools; http://sourceforge.net/projects/bbmap/) with the following parameters: qtrim = rl trimq = 25 minlen = 40. A two-pass STAR alignment (*Veeneman et al., 2016*) was performed against reference genomes with the same parameters as in our previous study (*Krizsán et al., 2019*; FPKM_calc.R) except that the maximal intron length was reduced to 3000 nt. Read count data was normalized using EdgeR (*Robinson et al., 2010*) as in our previous study (*Krizsán et al., 2019*). Expression levels were calculated as fragments per kilobase of transcript per million mapped reads (FPKM). Samples, such as FBCL and FBS of *C. cinerea from Krizsán et al., 2019*, and stage 2 primordia (P2) of *P. ostreatus,* were excluded from our analysis to avoid the signs of fruiting body autolysis and for quality reasons, respectively. Raw RNA-seq reads have been deposited to NCBI's GEO archive (GSE176181).

## Identification of developmentally expressed genes

Developmentally expressed genes were defined as genes that show at least twofold or fourfold change in expression between any two fruiting body stages or tissue types and that show an expression level FPKM > 4, as detailed in *Krizsán et al., 2019*. The gene was excluded if the maximum expression was detected in the VM.

## Species tree and relative gene age estimation

Protein sequences of 109 whole genomes (*Supplementary file 4a*) across Basidiomycota and Ascomycota (as outgroup) were downloaded from the JGI genome portal (September 2019; *Grigoriev et al., 2014*; *Nordberg et al., 2014*). All-vs-all similarity search was carried out with MMseqs2 (*Steinegger and Söding, 2017*) using three iterations and setting sensitivity to 5.7, max-seqs to 20,000, e-profile to 1e$^{-4}$, a preliminary coverage cutoff to 0.2, and an e-value cutoff to 0.001. Then, an asymmetrical coverage filtering was performed where we required ≥0.2 pairwise alignment coverage from the longer protein and ≥0.8 from the shorter one, with the aim to omit aspecific hits while retaining gene fragments (covercutter.R). Then, Markov clustering with an inflation parameter 2.0 was performed using the ratio of 'number of identical matches' (Nident) and 'query sequence length' (qlen) as weight

in the matrix. After clustering, we removed contaminating proteins from gene families following the logic of *Richter et al., 2018*.

For species tree reconstruction, we used 115 single-copy gene families that were present in ≥50% of the 109 species. Multiple sequence alignments were inferred using PRANK v.170427 (*Löytynoja, 2014*) and trimmed with TrimAL v.1.2 (-strict) (*Capella-Gutiérrez et al., 2009*). Trimmed MSA-s shorter than 100 amino acid (AA) residues were discarded. Best partitioning schemes, substitution models, and species tree reconstruction were performed under maximum likelihood (ML) in IQ-TREE v1.6.12 (*Minh et al., 2020*).

For gene tree reconstructions, gene families that contained at least four proteins were aligned with the MAFFT LINSI v7.313 (*Katoh and Standley, 2013*) algorithm or with FAMSA v1.5.12 (*Deorowicz et al., 2016*) and trimmed with TrimAL (gt-0.4). We inferred gene trees for each of the alignments in RAxmlHPC-PTHREADS-AVX2 8.2.12 under the PROTGAMMAWAG model of sequence evolution and assessed branch robustness using the SH-like support (*Stamatakis, 2014*). Rooting and gene tree/species tree reconciliation were performed with NOTUNG v2.9 (*Darby et al., 2017*) using an edge-weight threshold of 90. Then, a modified version of COMPARE (*Nagy et al., 2014*) was used to delineate orthogroups within gene trees.

Orthogroups were used to assign relative gene ages (hereafter: gene age), following standard phylostratigraphic definitions (*Domazet-Loso et al., 2007*): as the species tree node to which the MRCA of species represented in the orthogroup mapped. Enrichment of gene sets in gene age categories were analyzed with Fisher's exact test (R core team 2020).

## Transcriptome age index

TAI for each developmental stage of the nine investigated species was computed as described previously (*Domazet-Lošo and Tautz, 2010*) with slight modifications using the following formula: $TAI = \frac{\sum_{i=1}^{n} RA_i e_i}{\sum_{i=1}^{n} e_i}$ , where $RA_i$ represents the relative age of gene $i$, $e_i$ is the $\log_2$ FPKM value of gene $i$ at the given stage, and $n$ is the total number of genes. If available, tissue-specific expression values were averaged for each developmental stage. The TAI values of the investigated developmental stages were computed for each replicate, then averaged.

## Orthology based on reciprocal best hits

To characterize the conservation of developmental genes, we defined single-copy orthologs from the nine species based on reciprocal best hits between proteins. This strategy was stricter than the abovementioned orthogroup definition and was required to obtain functionally highly similar protein sets for comparing developmentally expressed genes. Proteins of each species were searched against the proteomes of other eight species using the RBH module of MMseqs2 with an e-value cutoff of 1e$^{-5}$. To remove spurious reciprocal best hits, we excluded a protein from the RBH group if its bit score was at least three times lower than the mean bit score of other hits of that query (self-hit excluded) and it shared <50% of its hits with those of the query (RBH_MMSeq.R). The orthogroups (one gene per species) obtained this way comprised considerably more focused gene sets than the approach used in *Krizsán et al., 2019*.

Orthogroups, which show developmentally dynamic expression with FC > 2 in at least four species, and with proportion ≥ 0.5, are considered hereafter as conserved developmental orthogroups. These conserved developmental orthogroups were also separated by the expression of *Cryptococcus* genes. We considered an orthogroup as 'shared orthogroup' if the *Cryptococcus* ortholog showed at least FC > 2 developmental expression, while we considered it as 'CM-specific' if the *Cryptococcus* ortholog was missing or did not show developmental regulation.

## Annotation of genes and gene families

We detected conserved domains in proteins using InterProScan-5.47–82.0 (*Jones et al., 2014*, IPRsimpcomp.R). Enrichment analysis on IPR domains was performed with Fisher's exact test (*R Development Core Team, 2020*), while enrichment analysis on GO categories was carried out using the R package topGO 2.44.0 (*Alexa and Rahnenfuhrer, 2016*). Proteins were further characterized by the best bidirectional hits to proteins of the model organisms *S. cerevisiae* (*Engel et al., 2014*), *Schizosaccharomyces pombe* (*Wood et al., 2012*), *Neurospora crassa* (*Galagan et al., 2003*), and *A. nidulans* (*Cerqueira et al., 2014*).

## RNA editing and allele-specific expression

To estimate the importance of RNA editing and ASE during fruiting body formation of *P. ostreatus*, we evaluated mismatches in Illumina reads according to their potential origin (RNA editing, ASE, noise). A custom pipeline (*Figure 10*) was constructed to first classify mismatches either as candidates for 'RNA editing'' or 'allele-specific.' Then, these mismatches were analyzed further in more specialized pipelines. First, we hard trimmed 10–10 nucleotides (nt) from both the 3′ and 5′ end of already quality trimmed reads to decrease the impact of sequencing errors during variant calling. A two-round STAR alignment was performed against both parental genomes (PC15 and PC9) as references, with the abovementioned parameters. Variants were identified with find_edit.awk script excluding bases with a Phred quality value below 30. Nucleotides differing the same way from both parental alleles were considered technical errors (caused by PCR amplification, sequencing, or alignment), somatic mutations, or RNA editing. Therefore, such mismatches were transferred to the RNA editing-specific pipeline. In contrast, variants that differed only from one of the parental genomes were attributed to ASE. The first part of the pipeline yielded the lists of variants that were further analyzed either in the RNA editing-specific pipeline or in ASE pipeline, as follows.

The RNA editing pipeline is detailed in Appendix 3 and *Figure 10*.

In the ASE pipeline (*Figure 10*), only previously assigned candidate allele-specific SNPs were considered. All reads were assigned to the parental genome to which it exhibited a smaller Hamming distance (Hd = number of SNPs). We assigned a read as indecisive if (i) Hd > 1 from both reference genomes, (ii) Hd > 15 from any of the reference genomes (too divergent read), or (iii) if the Hd was equal to both parental genomes. FPKM values were calculated as described above.

To describe the relative expression between the two parental nuclei (AS ratio), the number of PC15 reads was divided by the sum of parental-specific reads (PC15 + PC9) for each gene (g) in each sample (s): $AS = \frac{PC15_{gs}}{PC15_{gs} + PC9_{gs}}$ . An AS ratio close to 1 means dominant expression from the PC15 nucleus, whereas an AS ratio close to 0 means dominant expression from the PC9 nucleus. An AS ratio ~0.5 indicates equal expression from both nuclei. AS ratios were considered equal (set to 0.5) if (i) the expression was too low (FPKM < 2), (ii) the number of decisive reads was <16, and (iii) the proportion of indecisive reads was greater than 80%. We calculated two further measures, chromosome read ratio (CRR) and nuclear read ratio (NRR) introduced by *Gehrmann et al., 2018*, which represent the FPKM values of PC15 nucleus divided by the FPKM values of PC9 summed over chromosomes, and over all genes, respectively.

We identified genes with twofold (S2) and fourfold (S4) shifted expression between the two nuclei at AS cutoff values of AS < 0.31 or AS > 0.68 (corresponding to 5% quantile of all AS ratio values) and AS < 0.2 or AS > 0.8, respectively. For passing through, these filters and geometric means of replicates had to reach the upper limits (0.68 or 0.8) for PC15-specific ASE or less than lower limits (0.31 or 0.2) for PC9-specific ASE.

In order to understand how ASE may arise mechanistically, we compared putative promoters of ASE genes among parents, defined as the region from spanning 1000 nt upstream to 200 nt downstream of the transcription start site. Differences between the two parental regions were expressed as percent identity. For this analysis, the meanwhile released, improved version of PC9 was used (*Lee et al., 2021*). Protein sequences of parents were aligned with PRANK and an ML distance was calculated under WAG model (dist.ml function from phangorn R package; *Schliep, 2011*). Gene pairs with putative promoters <75% similar or where protein ML distances were >0.5 and alignment coverage <0.7 were removed to avoid potential orthology assignment errors. To identify the strength of selection for these genes, we inferred $\omega$ (dN/dS ratios) under two evolutionary models using CodeML, a program from PAML 4.4 (*Yang, 2007*). For this, CDSs from the genomes of species in the Pleurotinae clade (*Supplementary file 4a*) were extracted using GenomicFeatures packages (*Lawrence et al., 2013*). 1:1 orthologs were detected with MMseqs RBH function and codon aligned using –code option of PRANK. Reference tree for CodeML was extracted from the species tree, and $\omega$ values were calculated under 1-ratio (M0) model assuming that $\omega$ has been constant throughout the tree and free-ratio model (fb) allowing an independent $\omega$ for each branch in the tree. For statistical comparisons, the Kruskal–Wallis rank-sum test with Nemenyi post hoc test or paired Wilcoxon signed-rank test and Fisher's exact test were implemented in R (*R Development Core Team, 2020*).

## Acknowledgements

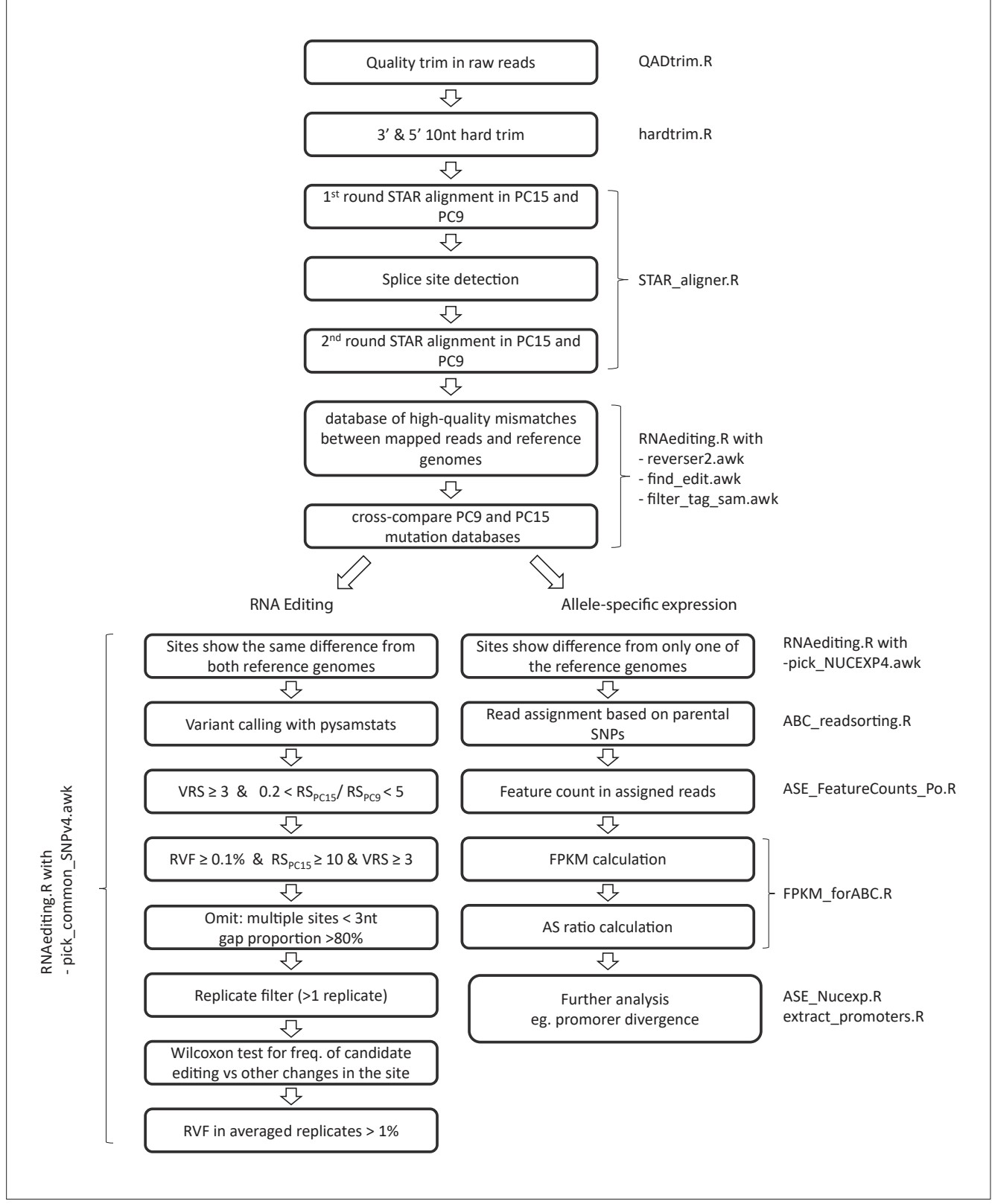

**Figure 10.** Pipeline of RNA editing and allele-specific expression annotation. Names of scripts available in Dryad (doi:https://doi.org.10.5061/dryad.5qfttdz5m) are displayed next to process boxes. VRS: variant read support; RS: read support; RVF: relative variant frequency.

We are thankful to Otto Miettinen for permission to utilize unpublished genomic data of *Aphanobasidium pseudotsugae*. We acknowledge support by the Hungarian National Research, Development, and Innovation Office (contract no. GINOP-2.3.2-15-2016-00052), the 'Momentum' program of the Hungarian Academy of Sciences (contract no. LP2019-13/2019 to LGN) and the European Research Council (grant no. 758161 to LGN).

## Additional information

### Funding

| Funder | Grant reference number | Author |
| --- | --- | --- |
| Hungarian National Research, Development, and Innovation Office | GINOP-2.3.2-15-2016-00052 | László G Nagy |
| Hungarian Academy of Sciences | Momentum Program LP2019-13/2019 | László G Nagy |
| European Research Council | 758161 | László G Nagy |

The funders had no role in study design, data collection and interpretation, or the decision to submit the work for publication.

### Author contributions

Zsolt Merényi, Conceptualization, Data curation, Formal analysis, Investigation, Resources, Visualization, Writing – original draft; Máté Virágh, Formal analysis, Investigation, Resources; Emile Gluck-Thaler, Conceptualization, Formal analysis, Investigation, Software, Validation, Visualization; Jason C Slot, Conceptualization, Formal analysis, Supervision; Brigitta Kiss, Investigation, Methodology, Resources; Torda Varga, Data curation, Formal analysis, Investigation, Visualization; András Geösel, Data curation, Investigation, Resources, Validation; Botond Hegedüs, Formal analysis, Methodology, Software; Balázs Bálint, Formal analysis, Investigation, Software, Validation; László G Nagy, Conceptualization, Formal analysis, Funding acquisition, Supervision, Visualization, Writing – original draft, Writing – review and editing

### Author ORCIDs

Zsolt Merényi ⓘ http://orcid.org/0000-0003-1114-3739
Jason C Slot ⓘ http://orcid.org/0000-0001-6731-3405
László G Nagy ⓘ http://orcid.org/0000-0002-4102-8566

### Decision letter and Author response

Decision letter https://doi.org/10.7554/eLife.71348.sa1
Author response https://doi.org/10.7554/eLife.71348.sa2

## Additional files

### Supplementary files

• Supplementary file 1. Literature-collected developmental genes in *Coprinopsis cinerea* and *Pleurotus* sp.

• Supplementary file 2. InterPro domain enrichment analysis for genes with allele-specific expression (S2 and S4) among all IPR annotated genes.

• Supplementary file 3. Conserved 1:1 ortholog groups.

• Supplementary file 4. Species that were used for transcriptomic analysis and for species tree.

• Supplementary file 5. Gene Ontology enrichment for complex multicellularity (CM)-specific and shared orthologs.

• Transparent reporting form

## Data availability

Raw RNA-Seq reads have been deposited to NCBI's GEO archive (GSE176181). Other data used in this study are available on Dryad (https://doi.org/10.5061/dryad.5qfttdz5m).

The following datasets were generated:

| Author(s) | Year | Dataset title | Dataset URL | Database and Identifier |
|---|---|---|---|---|
| Merényi Z, Virágh M, Gluck-Thaler E, Slot J, Kiss B, Varga T, Geösel A, Hegedüs B, Bálint B, Nagy L | 2021 | Data from: Gene age predicts the transcriptional landscape of sexual morphogenesis in multicellular fungi | http://dx.doi.org/10.5061/dryad.5qfttdz5m | Dryad Digital Repository, 10.5061/dryad.5qfttdz5m |
| Nagy LG | 2021 | Gene age predicts the transcriptional landscape of sexual morphogenesis in multicellular fungi | http://www.ncbi.nlm.nih.gov/geo/query/acc.cgi?acc=GSE176181 | NCBI Gene Expression Omnibus, GSE176181 |

The following previously published datasets were used:

| Author(s) | Year | Dataset title | Dataset URL | Database and Identifier |
|---|---|---|---|---|
| Krizsán K, Almási É, Merényi Z, Sahu N, Virágh M, Kószó T, Barry K | 2019 | Transcriptomic atlas of mushroom development reveals conserved genes behind complex multicellularity in fungi [Schizophyllum commune] | https://www.ncbi.nlm.nih.gov/geo/query/acc.cgi?acc=GSE125198 | NCBI Gene Expression Omnibus, GSE125198 |
| Krizsán K, Almási É, Merényi Z, Sahu N, Virágh M, Kószó T, Barry K | 2019 | Transcriptomic atlas of mushroom development reveals conserved genes behind complex multicellularity in fungi [Coprinopsis cinerea] | https://www.ncbi.nlm.nih.gov/geo/query/acc.cgi?acc=GSE125184 | NCBI Gene Expression Omnibus, GSE125184 |
| Krizsán K, Almási É, Merényi Z, Sahu N, Virágh M, Kószó T, Barry K | 2019 | Transcriptomic atlas of mushroom development reveals conserved genes behind complex multicellularity in fungi [Phanerochaete chrysosporium] | https://www.ncbi.nlm.nih.gov/geo/query/acc.cgi?acc=GSE125199 | NCBI Gene Expression Omnibus, GSE125199 |
| Liu L, He G J, Chen L, Zheng J, Chen Y, Shen L, Wang L | 2018 | Genetic basis for coordination of meiosis and sexual structure maturation in Cryptococcus neoformans | https://www.ncbi.nlm.nih.gov/geo/query/acc.cgi?acc=GSE111975 | NCBI Gene Expression Omnibus, GSE111975 |
| Almasi E, Sahu N, Krizsan K, Balint B, Kovacs G M, Kiss B, Chovatia M | 2019 | Comparative genomics reveals unique wood-decay strategies and fruiting body development in the Schizophyllaceae | https://www.ncbi.nlm.nih.gov/geo/query/acc.cgi?acc=GSE132826 | NCBI Gene Expression Omnibus, GSE132826 |
| Sipos G, Prasanna A.N, Walter M.C, O'Connor E, Balint B, Krizsan K | 2017 | Lineage-specific genetic innovations streamline the genomes of Armillaria species to pathogenesis | https://www.ncbi.nlm.nih.gov/geo/query/acc.cgi?acc=GSE100213 | NCBI Gene Expression Omnibus, GSE100213 |
| Gehrmann T, Pelkmans JF, Ohm RA, Vos AM, Sonnenberg AS, Baars JJ | 2018 | Nucleus-specific expression in the multinuclear mushroom-forming fungus Agaricus bisporus reveals different nuclear regulatory programs. | https://www.ncbi.nlm.nih.gov/bioproject/PRJNA309475 | NCBI BioProject, PRJNA309475 |
| Ke HM, Lee HH, Lim CYI, Liu YC, Lu MR, Hsieh JWA | 2020 | Mycena genomes resolve the evolution of fungal bioluminescence. | https://www.ncbi.nlm.nih.gov/bioproject/?term=PRJNA623720 | NCBI BioProject, PRJNA623720 |

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

# Appendix 1

## Identification of gene clusters

Genes contributing to the same phenotype can occasionally be found clustered together in fungal genomes. Because such clustering is predicted to both facilitate gene co-expression and maintain linkage over long periods of evolutionary time, we tested whether any developmentally expressed genes were physically clustered together more often than expected by chance, and whether any of these clusters were conserved across species. Furthermore, since many secondary metabolites encoded by gene clusters play a role in development, we tested if any of these developmentally expressed gene clusters might overlap with predicted biosynthetic gene clusters in eight Basidiomycota species, including *P. ostreatus* and *Pt. gracilis* (Dryad: Table D2). We first designated all developmentally expressed genes separated by six or less intervening genes as candidate clusters. We then calculated the probability of observing each cluster using a binomial test, where expected cluster size was estimated assuming that developmentally expressed genes are randomly distributed in the genome. Clusters with a probability of observation <0.01 were designated clustering 'hotspots.' We checked hotspots for overlap with predicted biosynthetic gene clusters using de novo antiSMASH v5 annotations (*Medema et al., 2011*). Conservation of hotspots between different fungal species was assessed by BLASTp (*Altschul et al., 1990*) using two metrics: percent gene content similarity, which is the number of genes in the query hotspot that are also clustered together in the target genome; and percent FDBR similarity, which is the number of developmentally expressed genes in the query hotspot that are also developmentally expressed and clustered together in the target genome.

## Developmental gene clusters are not conserved

Having established that the developmental hourglass may not apply to fungi, we next asked if we can find evidence in fungi for the physical clustering of developmental genes in the genome, a characteristic of several key genes involved in animal pattern formation. Certain fungal genes, such as those encoding secondary metabolite biosynthetic pathways, are well known to cluster physically (*Keller, 2019*), whereas similar traits for developmental genes have not yet been investigated. We found evidence for the occasional grouping of developmentally expressed genes into hotspots in the genomes (see Materials and methods; *Appendix 1—figure 1*, Dryad: Table D2). Altogether 153 hotspots were detected in eight genomes; however, most of these were species-specific and not conserved across species (e.g., the luciferase cluster in the bioluminescent *A. ostoyae*; *Ke et al., 2020*). Surprisingly, most hotspots did not overlap with predicted biosynthetic gene clusters (*Appendix 1—figure 2*). The presence of hotspots but the lack of conservation suggests that these developmental gene clusters may be linked to species-specific developmental traits.

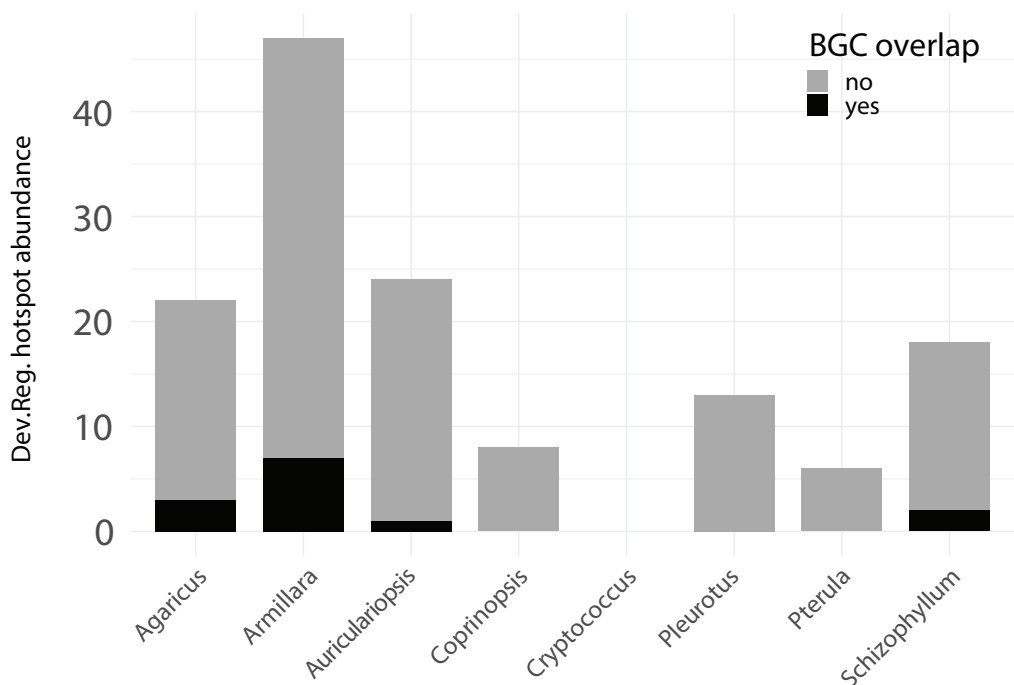

**Appendix 1—figure 1.** Developmentally expressed genes occasionally cluster together in genomic 'hotspots.'. A bar chart summarizing the number of hotspots detected per genome (total = 153) and the degree of overlap between hotspots and predicted biosynthetic gene clusters (BGCs).

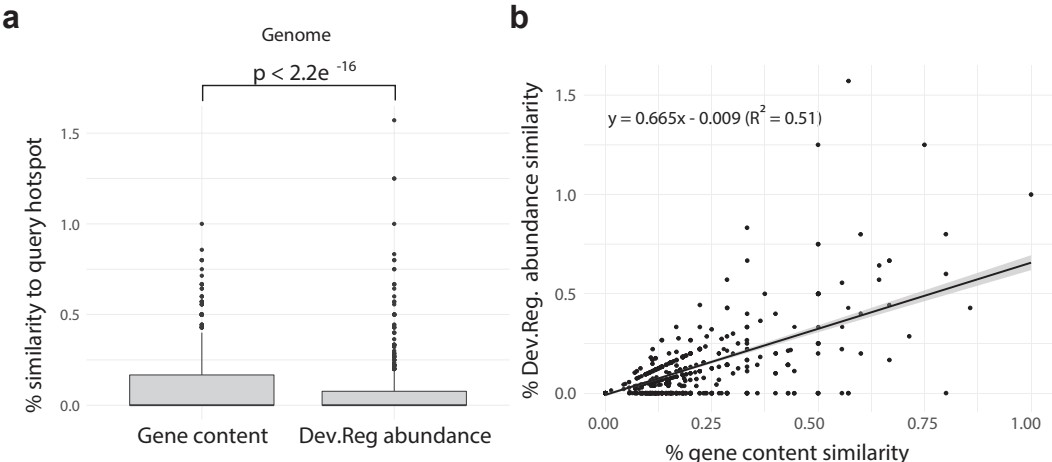

**Appendix 1—figure 2.** Developmentally expressed genes occasionally cluster together in genomic 'hotspots.'. (**a**) A box-and-whisker plot summarizing the distribution of % gene content conservation and the distribution of % Dev.Reg. abundance conservation of all 153 hotspots when searched for in genomes other than the one in which they are found. Significance between distributions determined by the Wilcoxon rank-sum test. (**b**) Scatterplot and linear regression describing the relationship between the % gene content conservation and % Dev.Reg. abundance conservation of each of the 153 hotspots when searched for in genomes other than the one in which they are found (number of observations = 153 hotspot queries × 8 target genomes). % Dev.Reg. abundance conservation may exceed 100% if more Dev.Reg. genes are found in target regions as compared with query hotspot. The shaded region around the fitted regression line represents 95% confidence interval.

## Appendix 2

### Identification of natural antisense transcripts

NATs were defined as de novo-assembled transcripts located antisense to a gene >200 nt long, not showing similarity to structural RNA species, not overlapping with UTRs of neighboring genes, and showing an expression above a given cutoff (*Appendix 2—figure 1*). For de novo transcript assembly, quality filtered reads were first mapped to the reference genome using STAR_2.6.1a_08–27 (*Veeneman et al., 2016*). After identifying splice sites, a second mapping was performed. StringTie version 2.0.3 (*Pertea et al., 2015*) was used to generate a genome-guided de novo transcriptome assembly and annotation (NATextractor.R). Transcripts shorter than 200 nt or supported by <5 reads in a single sample were excluded. Output GTF files from each sample were merged, compared to the reference annotation with gffcompare v0.11.2 (*Pertea and Pertea, 2020*), and transcripts with exonic overlap on the opposite strand (i.e., antisense; class_code x) were retained. To exclude transcripts mapped to repeats, RepeatModeler v2.0 (*Flynn et al., 2020*) was used to identify repeat regions of genomes. Conserved structural RNA transcripts (tRNAs, U2 spliceosome, ribosomal RNAs, Hammerhead ribozymes) were identified with Infernal 1.1.3 (*Nawrocki, 2013*) based on the Rfam database (*Kalvari et al., 2021*) and were removed. Fungal genomes are densely packed with genes, raising the possibility that a detected transcript is actually the UTR region of the closest gene (*Rhind et al., 2011*). To avoid identifying UTR regions as NATs, candidate NATs showing a strongly correlated expression (Pearson's p-value < 0.05) and located <500 nt from the closest coding genes on the same strand were discarded from further analysis. Coding potential of NATs was characterized based on the default cutoff of the Coding Potential Calculator CPC2 (*Kang et al., 2017*).

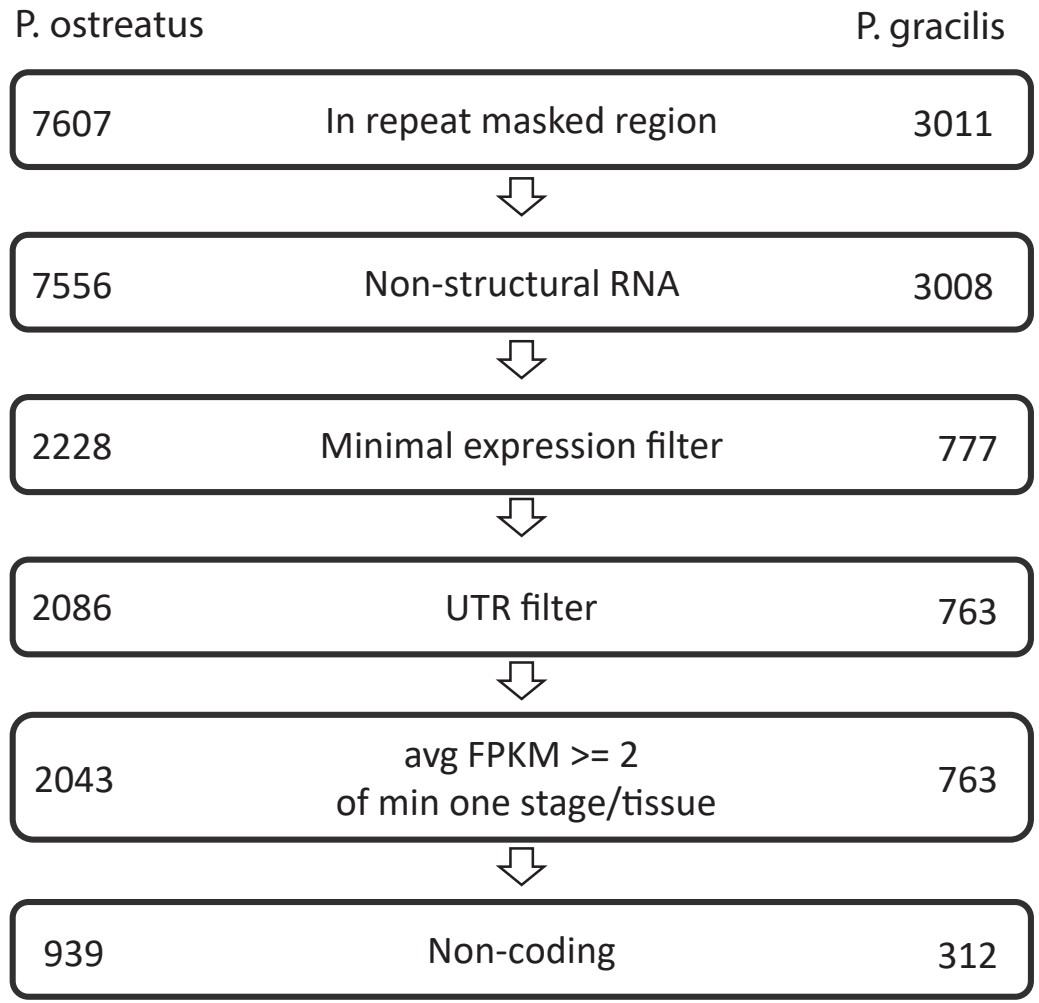

**Appendix 2—figure 1.** Pipeline of natural antisense transcripts. Numbers represent the retained transcripts in each filtering step.

Expression level of NATs was quantified with the FeatureCounts R package (*Liao et al., 2014*; NAT_FeatureCounts.R) based on the union of the exons per transcript. FPKM calculations were carried out as mentioned above, only transcripts with at least five mapped reads in at least three samples were retained. Developmental regulation of potential NATs was calculated as for genes. To assess conservation of NATs, we mapped the transcripts on the genomes of the 109 species with minimap2 v 2.17 (options: -k15 -w5 `--splice -g2000 -G200k` -A1 -B1 -O1,20 -E1,0 -C9 -z500 -ub `--junc-bonus` = 9 `--splice-flank` = yes) (*Li, 2018*).

### Natural antisense transcripts show fast turnover

NATs are abundantly transcribed from fungal genomes and can include important regulatory RNAs that influence, among others, sexual development (*Donaldson et al., 2017*; *Donaldson and Saville, 2012*; *Faghihi and Wahlestedt, 2009*; *Kim et al., 2018*). We analyzed NATs in *P. ostreatus* and *Pt. gracilis* based on strand-specific RNA-seq data and identified 2043 and 763 de novo transcripts as NATs (*Appendix 2—figure 1*), corresponding to 17.6 and 6.3% of protein coding genes, respectively. Lengths, exon structures, and coding potentials of the assembled NATs were similar to those in earlier reports (*Borgognone et al., 2019*; *Kim et al., 2018*; *Wang et al., 2019*; *Appendix 2—figures 1–2*).

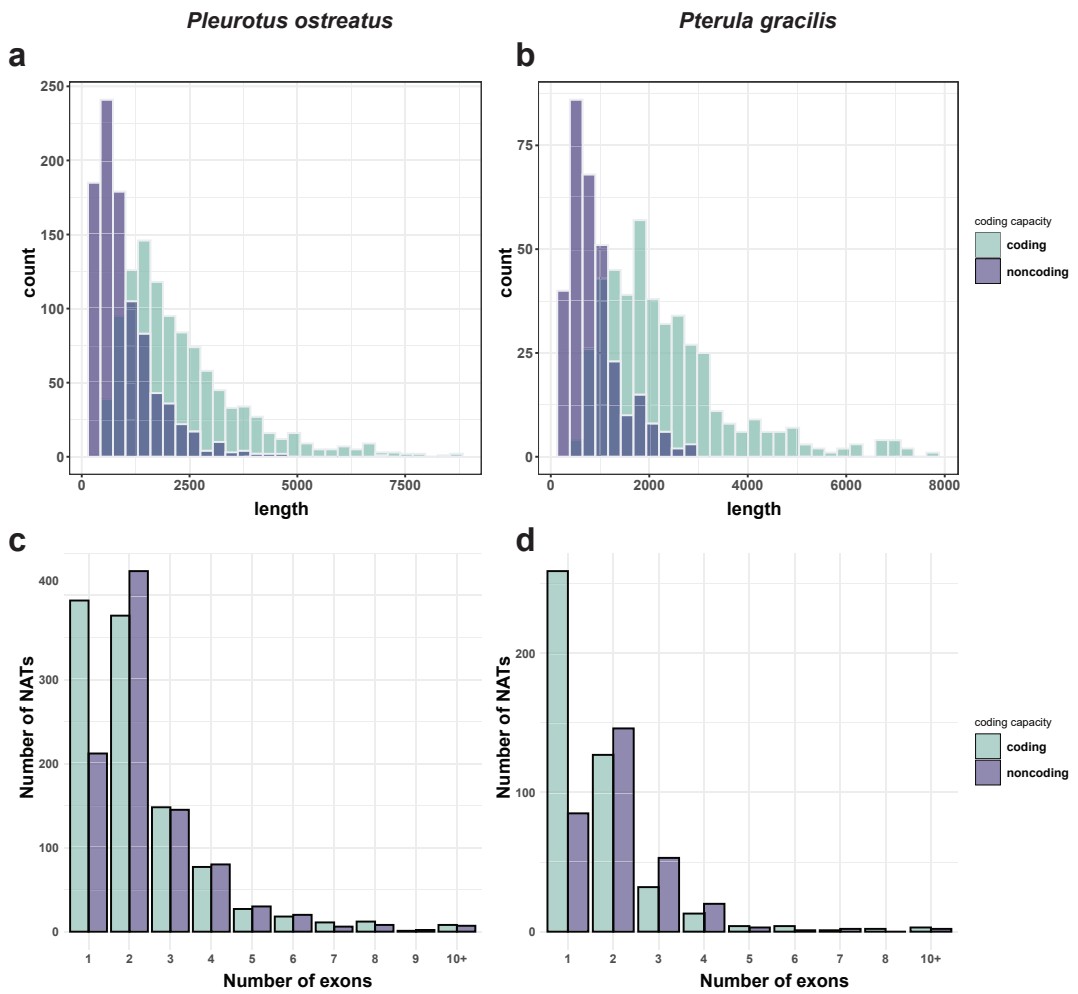

**Appendix 2—figure 2.** Length (**a, b**) and exon number (**c, d**) distribution of natural antisense transcripts (NATs) in *Pleurotus ostreatus* (**a, c**) and *Pterula gracilis* (**b, d**). NATs were divided into putatively coding and noncoding categories using the default settings of CPC2 (**Kang et al., 2017**).

NATs showed developmentally dynamic expressions with NAT expression patterns reflecting stage and tissue identity, with tight grouping of biological replicates (**Appendix 2—figure 3**). Similarly to the pattern of MDS for total gene expression, FBC and FBS samples and the early stages (P1, P3, and YFB) were close to each other. As many as 1173 (57.4%) and 126 (16.5%) NATs of *P. ostreatus* and *Pt. gracilis* were developmentally expressed, respectively. These may expand the space of developmentally expressed transcripts in fruiting bodies, thus, irrespective of their exact mechanism of action, can contribute to CM. In *P. ostreatus*, we identified 166 NATs (8.1%) that showed at least a twofold higher expression in all fruiting body stages than in VM, comparatively more than predicted coding genes (4.8%). Such transcripts may regulate the transition from simple multicellularity in VM to complex multicellularity in FB, one of the most significant transcriptomic reprogramming events in the fungal life cycle (**Krizsán et al., 2019**). **Kim et al., 2018** found a considerable proportion (21.3%) of lncRNA (which overlap only partially with NATs) that might have a role in sexual development of *Fusarium graminearum*. Nevertheless, only 39.1% of *P. ostreatus* and 4.1% of *Pt. gracilis* of the NAT-possessing genes show developmental regulation with at least a fourfold change.

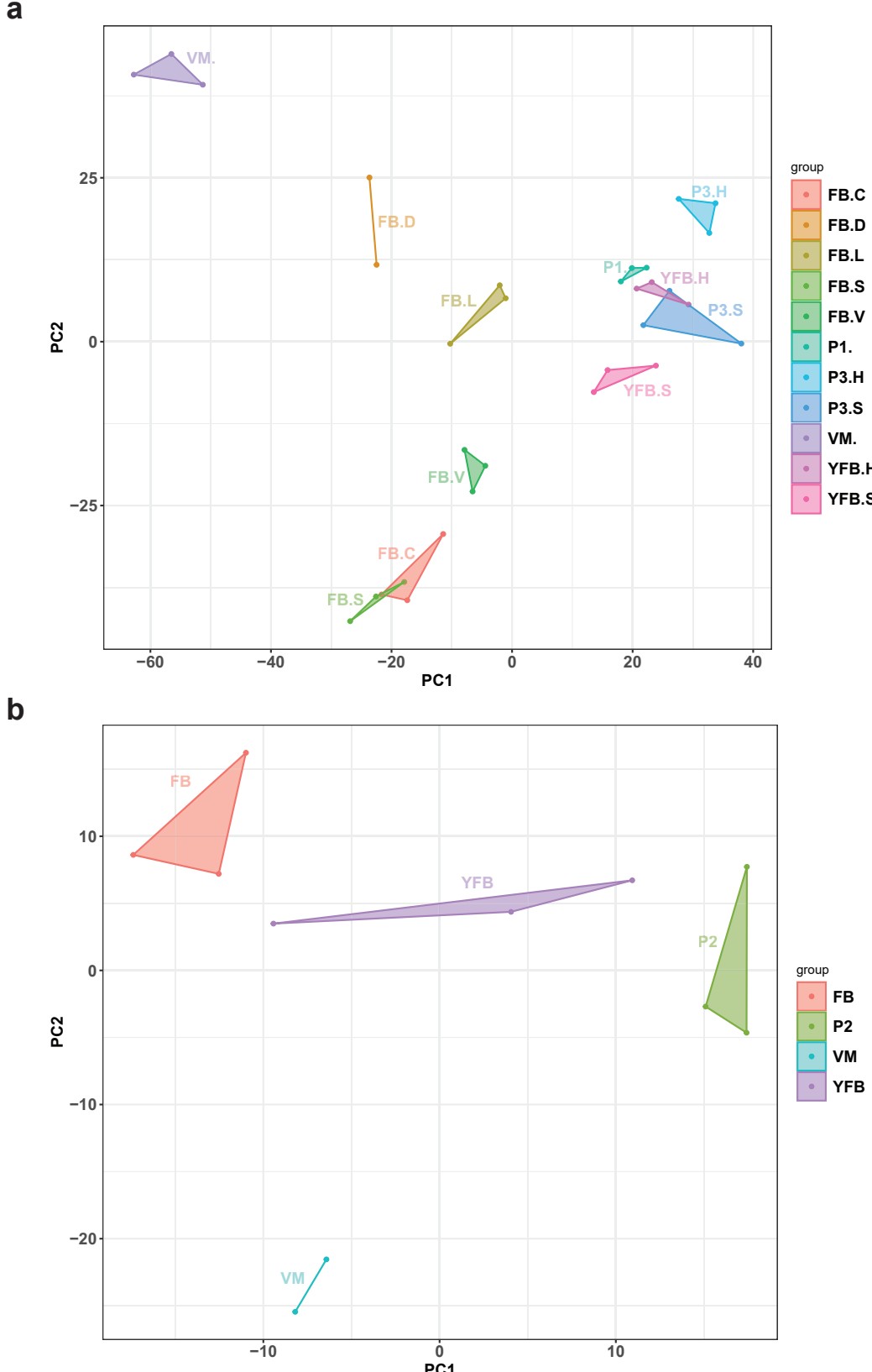

**Appendix 2—figure 3.** Principal component analysis (PCA) plot based on the expression of natural antisense transcripts detected in (**a**) *Pleurotus ostreatus* and (**b**) *Pterula gracilis*. VM: vegetative mycelium; P1: stage 1 *Appendix 2—figure 3 continued on next page*

primordium; P3: stage 3 primordium; YFB: young fruiting body; FB: fruiting body; H: cap (entire); C: cap trama (only the inner part, without lamellae, or skin); L: lamellae; S: stipe; V: cuticle; D: dedifferentiated tissue of cap.

The detected NATs displayed low-sequence conservation. Of the 2043 NATs of *P. ostreatus*, 1815 (89%) showed homology in the closest sequenced species (*Pleurotus eryngii*) and only 177 (8.7%) in other species (*Appendix 2—figure 4*). In *Pt. gracilis*, only 15 (2.0 %) NATs showed homology in other species (*Appendix 2—figure 5*). We find that low overlap with gene exons can explain the lack of sequence conservation in NATs. In *P. ostreatus*, only 596 of the 2043 NATs (29%) showed at least 75% total exon–exon overlap. This suggests that, even if located in conserved genes, NATs mostly overlap with introns or intergenic regions, which may allow rapid sequence turnover, as reported in vertebrates (*Kapusta and Feschotte, 2014*). We also failed to detect conservation of sense gene identity: of the 6232 co-orthologous genes between *P. ostreatus* and *Pt. gracilis*, only 70 showed evidence for NAT in both species. All of these observations imply low conservation of NATs at the sequence level, consistent with the view that NAT homology is detectable only among closely related species (*Donaldson and Saville, 2013*; *Rhind et al., 2011*).

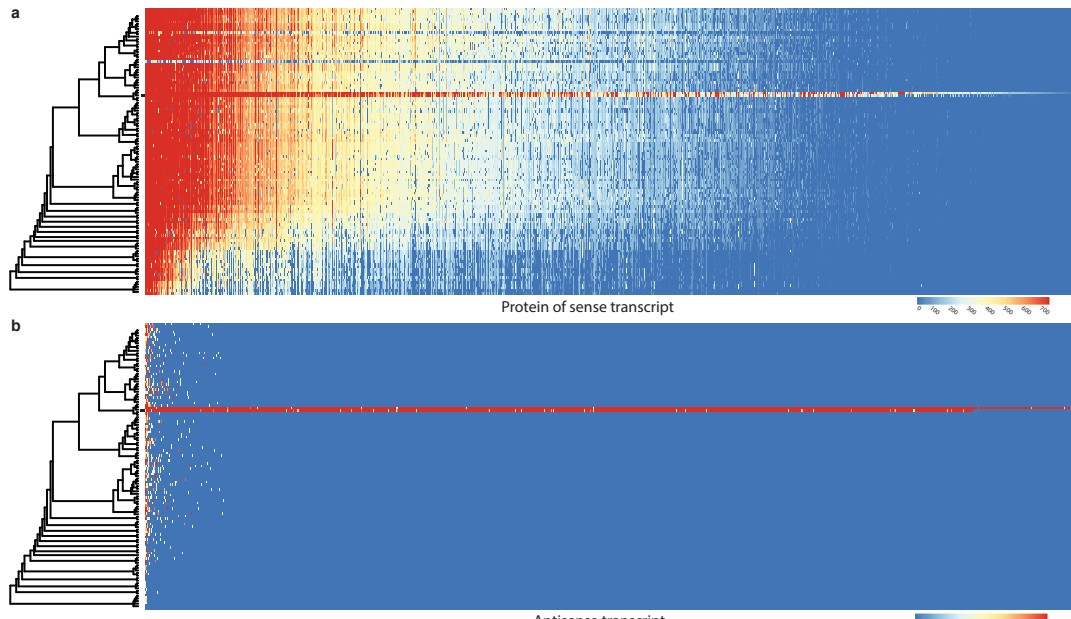

**Appendix 2—figure 4.** Conservation of sense genes and their natural antisense transcripts (NATs) of *Pleurotus ostreatus* across 109 species. (**a**) Similarity of proteins of sense transcripts – having antisense transcripts – measured with –log₁₀(e-value) from MMseqs search against the 109 species dataset. (**b**) Mapping score of antisense transcripts based on minimap2. Warmer color represents a higher similarity according to the scales. Black square denotes *P. ostreatus*. Rows represent the species while columns represent the antisense query transcripts (**b**) or proteins from sense transcripts (**a**). For a larger species tree, see *Figure 1*.

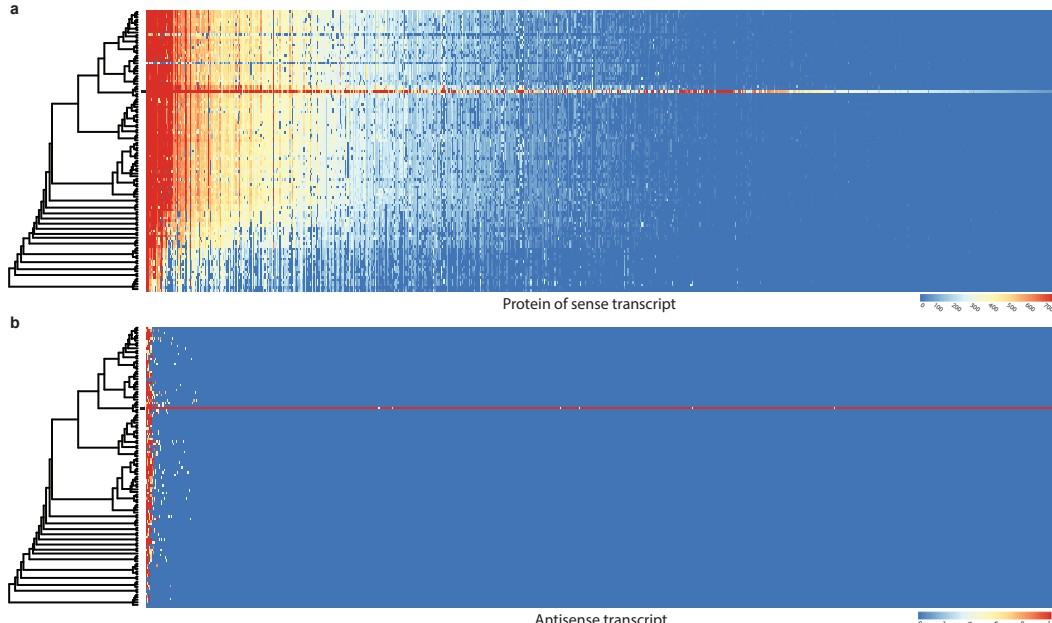

**Appendix 2—figure 5.** Conservation of sense genes and their natural antisense transcripts (NATs) of *Pterula gracilis* across 109 species. (**a**) Similarity of proteins of sense transcripts – having antisense transcripts – measured with –log$_{10}$(e-value) from MMseqs search against the 109 species dataset. (**b**) Mapping score of antisense transcripts based on minimap2. Warmer color represents a higher similarity according to the scales. Black square denotes *Pt. gracilis*. Rows represent the species while columns represent the antisense query transcripts (**b**) or proteins from sense transcripts (**a**). For a larger species tree, see *Figure 1*.

In *P. ostreatus*, 263 NATs (12.8%) showed significant positive (Pearson *r* > 0.7, p<0.05, *Appendix 2—figures 6–7*, p<0.01) while 33 showed significant negative expression correlation (Pearson *r* < –0.7, p<0.05, *Appendix 2—figure 8*) with their sense genes. An enrichment of positive over negative correlation between sense and antisense transcript pairs was noted previously in *Fusarium* and in *Ganoderma lucidum* (*Kim et al., 2018*; *Shao et al., 2017*). Positively correlating pairs may be co-regulated via chromatin accessibility or be stabilized via dsRNA formation (*Donaldson and Saville, 2013*), whereas negative correlation can be explained by transcriptional interference, antisense-mediated chromatin remodeling or RNA masking (reviewed in *Donaldson and Saville, 2012*), all of which may be relevant to CM, but further research is needed to clarify their roles.

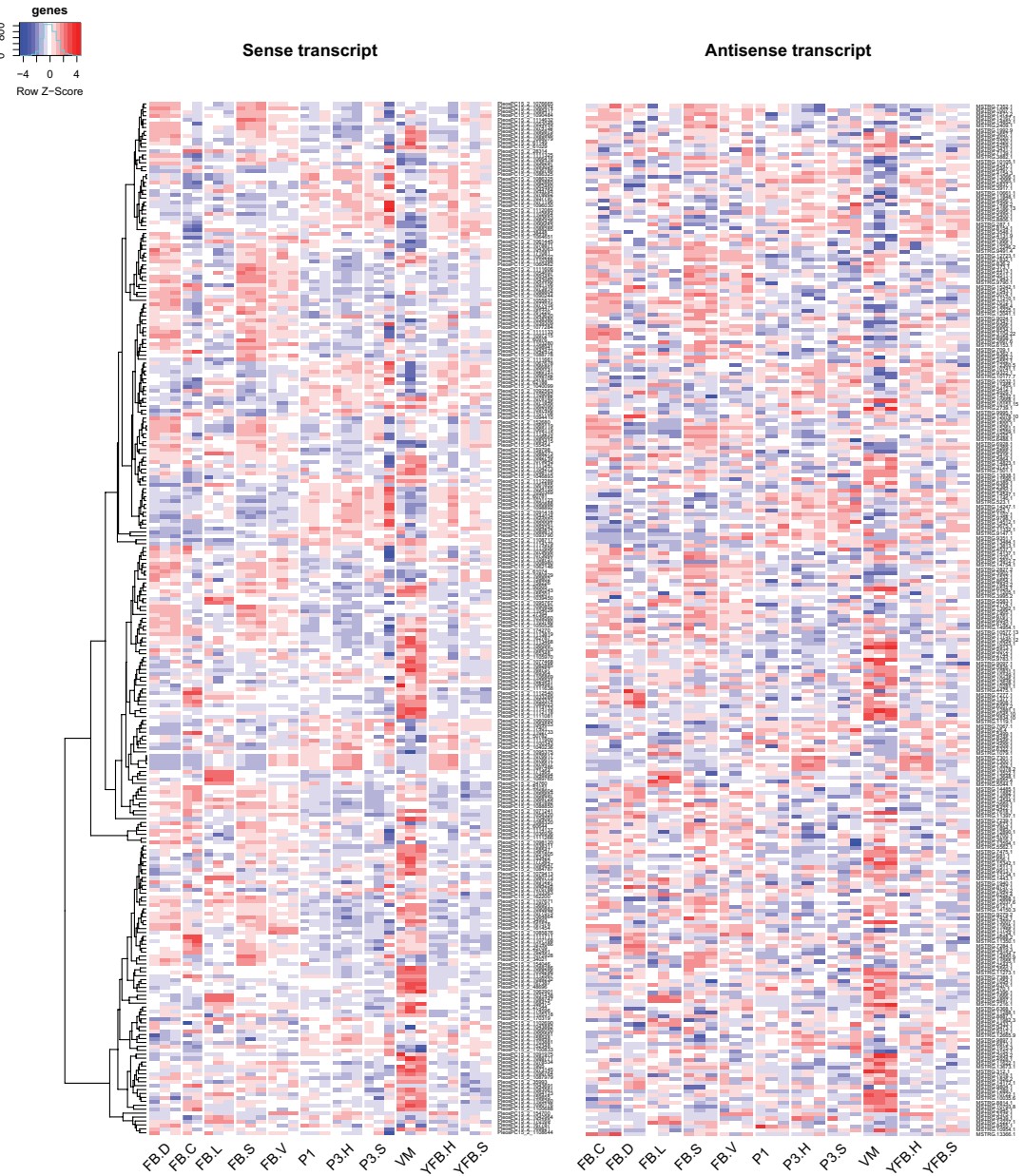

**Appendix 2—figure 6.** Expression pattern of 263 sense transcripts and their antisense transcripts that showed significant positive correlation (Pearson $r \geq 0.7$, $p<0.05$) in *Pleurotus ostreatus*. Corresponding lines of the heatmaps contain sense and antisense transcript pairs. VM: vegetative mycelium; P1: stage 1 primordium; P3: stage 3 primordium; YFB: young fruiting body; FB: fruiting body; H: cap (entire); C: cap trama (only the inner part, without lamellae, or skin); L: lamellae; S: stipe; V: cuticle; D: dedifferentiated tissue of cap.

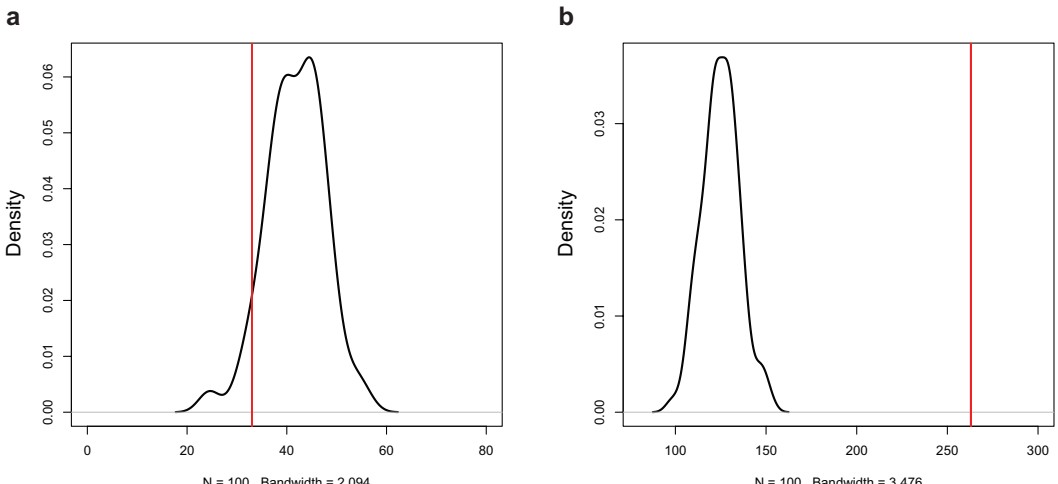

**Appendix 2—figure 7.** Permutation test for the number of (**a**) negative ($r < -0.7$, p<0.05) and (**b**) positive ($r > 0.7$, p<0.05) correlations among the expression of natural antisense transcripts (NATs) and random genes. Red line represents the observed number of significant correlations.

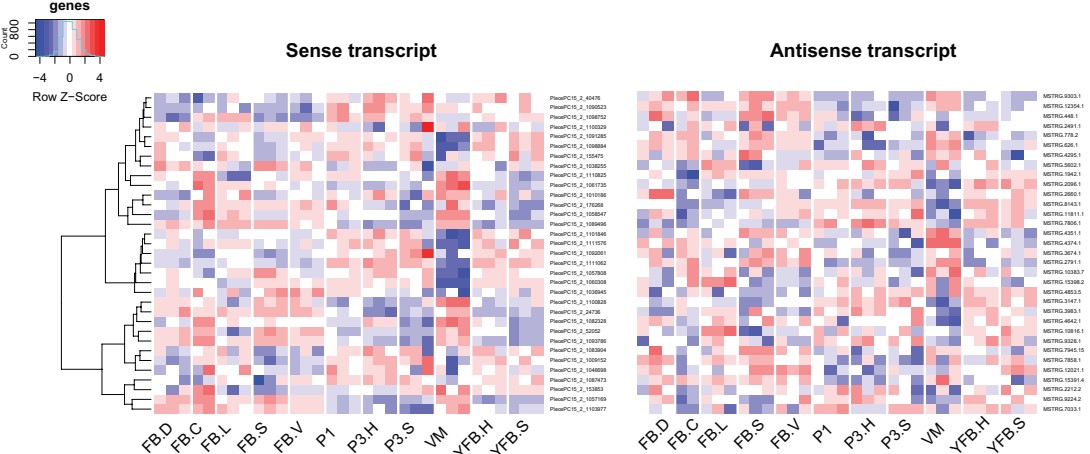

**Appendix 2—figure 8.** Expression pattern of 33 sense transcripts and their antisense transcripts that showed significant negative correlation (Pearson $r \le -0.7$, p<0.05) in *Pleurotus ostreatus*. Corresponding lines of the heatmaps contain sense and antisense transcript pairs. VM: vegetative mycelium; P1: stage 1 primordium; P3: stage 3 primordium; YFB: young fruiting body; FB: fruiting body; H: cap (entire); C: cap trama (only the inner part, without lamellae, or skin); L: lamellae; S: stipe; V: cuticle; D: dedifferentiated tissue of cap.

Together, the developmentally relevant expression, the lack of functional clues (*Appendix 2— figure 9*) and the low conservation of NATs suggest that antisense transcription is a fast-evolving component of CM transcriptomes with potential functions in modulating gene expression. Nevertheless, as above, nonadaptive explanations should not be ruled out, such as some NATs being transcriptional noise or leakage (*Dahary et al., 2005*; *Lloréns-Rico et al., 2016*).

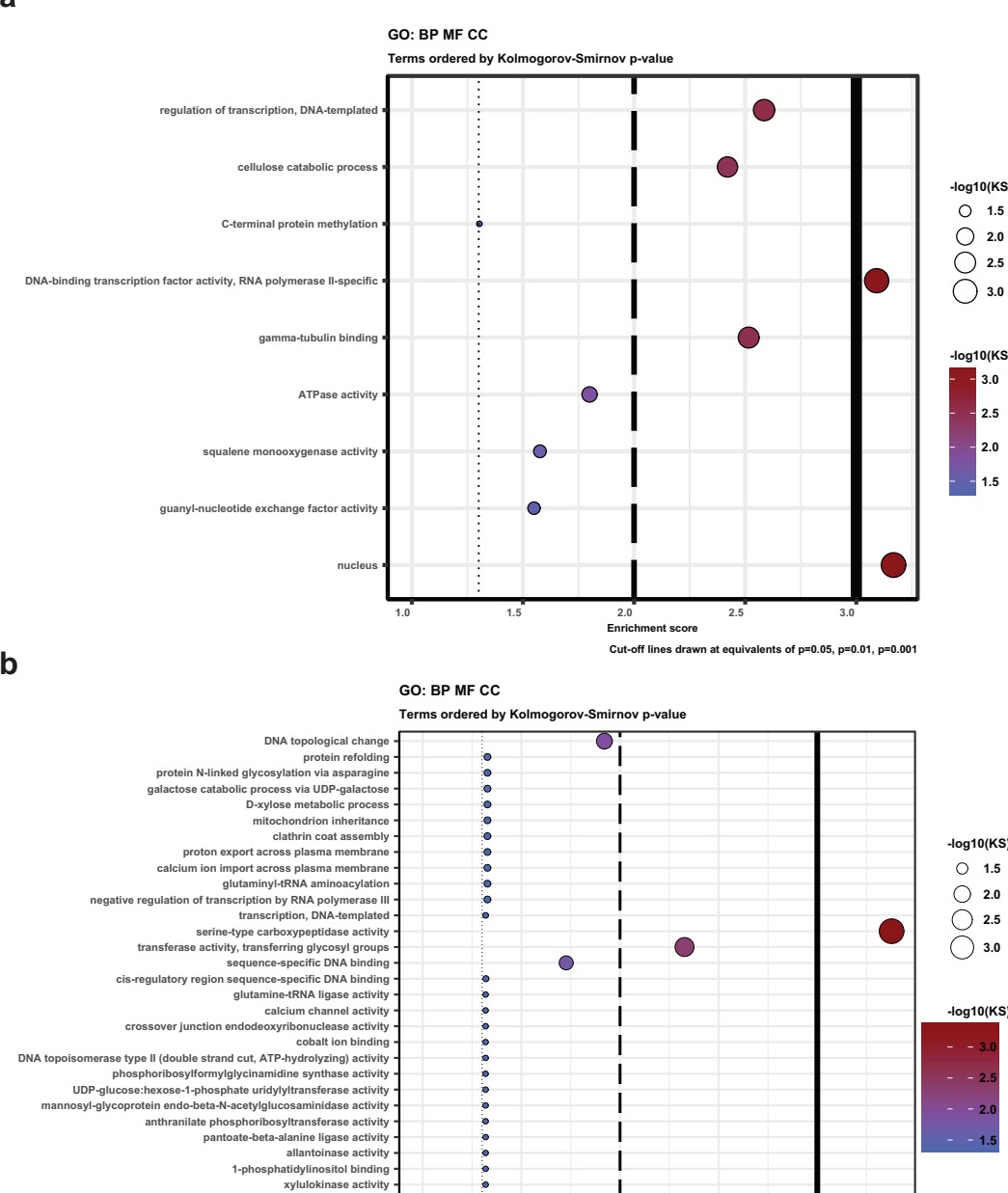

**Appendix 2—figure 9.** Gene Ontology (GO) enrichment for genes that have natural antisense transcripts (**a**) in *Pleurotus ostreatus* and (**b**) *Pterula gracilis*. KS means the p-value of Kolmogorov–Smirnov test implemented in the R package 'topGO.' BP: biological process; MF: molecular function; CC: cellular component.

## Appendix 3

### RNA editing pipeline

In the RNA editing pipeline (*Figure 10*), we re-called variants for sites with pysamstats 1.0.1 (https://github.com/alimanfoo/pysamstats; *Miles, 2017*; min-base quality 30 and max depth to 500,000) that were input to the RNA editing pipeline. We continued the analysis with only those variants that had ≥3 supporting reads, mapped to both reference genomes, and the total read support did not differ more than five times between the two parental mappings in order to avoid signal coming purely from the differential mapping to the two reference genomes (e.g., erroneous alignment). Further, we removed variants where read coverage was <10, a single variant was supported by <3 reads, or the proportion of the variant was below 0.1% in order to reduce the effect of technical errors, but retain editing sites. Because erroneous alignment around splice sites can produce variants indistinguishable from editing events, we discarded variants in which multiple sites with mismatches grouped within 3 nt distance of each other and in which the proportion of gaps exceeded 80% of the read coverage. After this step, we kept only variants that were present in at least two biological replicates. In addition, for a variant to be considered an RNA editing site, it had to be significantly more frequent across all samples (Wilcoxon rank-sum test with p<0.01) than any other nucleotide at that site (except reference). Finally, we considered a site an RNA editing site if the geometric mean of its frequencies across the three replicates exceeded 1%. Relative to the editing site, −3 upstream and +3 downstream surrounding sequences were extracted with rtracklayer package (*Lawrence et al., 2009*) and motifs were searched with the seqlogo package (*Bembom and Ivanek, 2020*).

### RNA editing is not abundant in fruiting body transcriptomes

In the RNA editing pipeline (*Figure 10*), 627,093 of the 1,999,221 input variants remained after filtering for extreme low frequency (<0.1%). We chose this permissive threshold (as opposed to 1/3/10% in other protocols; *Zhu et al., 2014*) to avoid discarding any signal in the early steps. As many as 546,790 sites were located in gene regions, of which 346,105 possibly corresponded to erroneous mapping around splice sites (*Appendix 3—figure 1*), while 218,685 were retained for further analysis. Surprisingly, only 1.2% of these (2701 sites) were consistent between at least two of the three biological replicates. After eliminating potential sequencing errors (Wilcoxon signed-rank test, p-value<0.01), we obtained 1179 variants. Requiring at least 1% mean variant frequency in at least one stage left 332 potential RNA editing sites, with 6–62 in each variant type (Dryad: Table D5). Among these, A-to-I and C-to-U transitions were not enriched (*Appendix 3—figure 2a*), consistent with previous Basidiomycota studies (*Bian et al., 2019*; *Teichert, 2020*). To explore other explanations, we examined what, other than RNA editing, our remaining variants could potentially correspond to. By examining motifs around the 332 sites, we detected solely an enrichment of adenines 1–2 nt downstream of C-to-A sites (*Appendix 3—figure 2c*). However, because 69% (20 of 29) of these were within ±20 nt from the 3′ end of the last exon of genes, we think that adenine enrichment corresponds to the polyadenylation sequence. Together, we interpret these results as limited or no evidence for RNA editing in *P. ostreatus*.

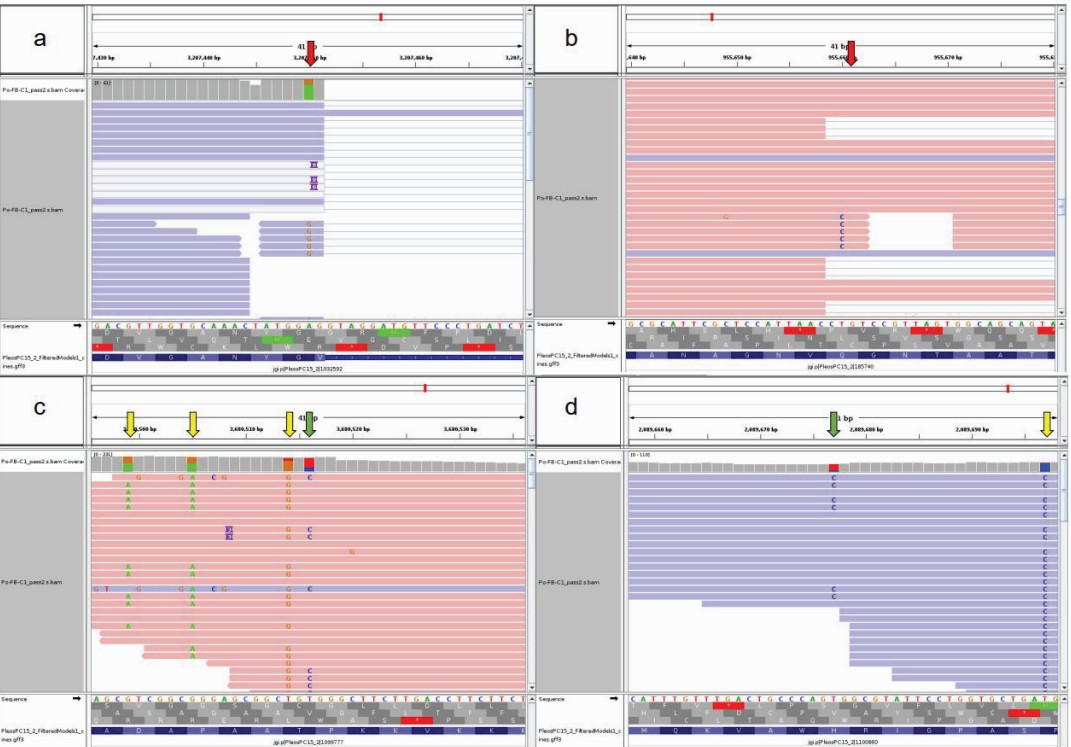

**Appendix 3—figure 1.** Examples for variants of different types. In (**a**) and (**b**), erroneous read alignment around splice sites causing variants (red arrows) similar to RNA editing. In (**c**) and (**d**), green arrows represent potential RNA editing sites, while yellow arrows represent allele-specific SNPs.

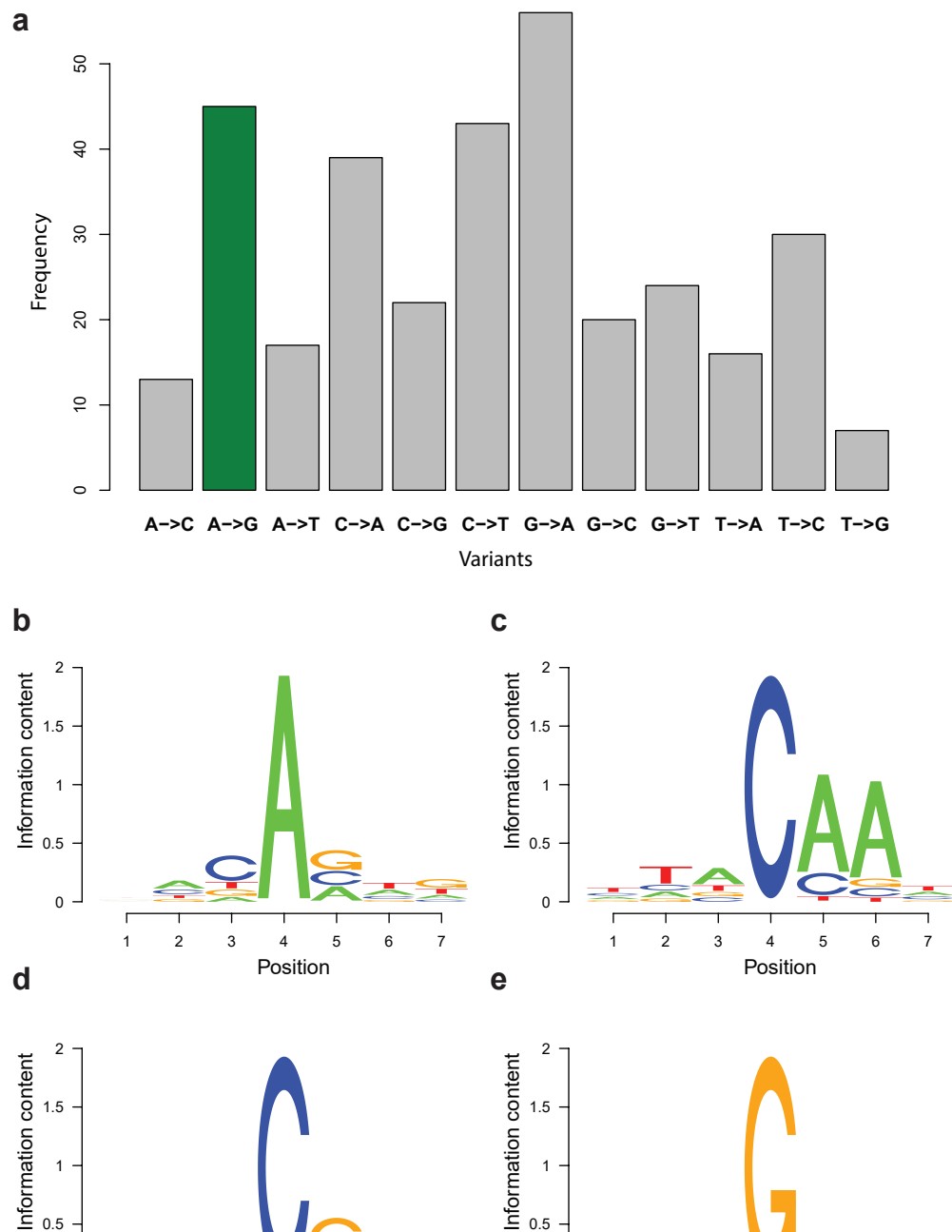

**Appendix 3—figure 2.** Enrichment of variant types and motifs among potential RNA editing sites. (**a**) Distribution of variant types retained in the RNA editing-specific pipeline with A-to-I variants being marked with green. (**b–e**) Sequence motifs surrounding the most frequent candidate RNA editing changes displayed as sequence logos. Fourth position represents the variants among reads. 1–3 is the upstream three positions, while 4–7 is the downstream three positions. (**b**) A-to-G, (**c**) C-to-A, (**d**) C-to-T, and (**e**) G-to-A changes.

