## [Editor Report]

This study sought to systematically identify key aspects of the transcriptional landscape in fungi that exhibit complex multicellularity (CM), associated with fruiting bodies. The authors examined a series of parameters of expression signatures, concluding that the best predictor of a gene behavior in the CM transcriptome was evolutionary age. Thus, the expression pattern of fruiting bodies showed a distinct gene age-related stratification, where it was possible to sort out genes related to general sexual processes from those likely linked to morphogenetic aspects of the CM fruiting bodies. Notably, these results do not support a developmental hourglass concept, which is the rather predominant hypothesis in metazoans, including some analysis in fungi.

---

## [Decision Letter]

**Decision letter after peer review:**

Thank you for submitting your article "Gene age predicts the transcriptional landscape of sexual morphogenesis in multicellular fungi" for consideration by *eLife*. Your article has been reviewed by 4 peer reviewers, including Luis F Larrondo as the Reviewing Editor and Reviewer #1, and the evaluation has been overseen by Patricia Wittkopp as the Senior Editor. The following individual involved in review of your submission has agreed to reveal their identity: Jason Stajich (Reviewer #4).

Essential revisions:

1) The manuscript needs to better integrate and streamline the analyses to clarify the paper's main question. For example, as indicated by REV#2, a clear message about what types of transcripts are linked to multicellularity is missing: instead of just mentioning DEGs in critical stages, more context is needed regarding the ones playing a key role in development and how they are regulated.

2) An in-depth analysis of the most relevant genes and pathways is necessary. While one could think of experimental validation (i.e. knocking out/ overexpressing candidate genes), taking in consideration that the paper has a main evolutionary context, a more in-depth analysis of genes/regulatory motifs/pathways and a better description of the associated gene programs could be an alternative. Likewise, they need to better elaborate some connections between sections of the manuscript (i.e. hourglass model and ASE), and provide case examples for other key aspects of the manuscript (i.e point 4 by REV#3)

3) The reviewers suggested that some of the data should be removed from the manuscript as it does not clearly contribute to the main point that is trying to make. This also applies to supplementary data. Moreover, they indicated that there is a very large number of supplementary files (tables and figures), where the presented data is not always clearly described. The authors should more finely curate what is included, removing data that is not essential.

4) Several of the sections need to be restructured (paying attention on how/when Figure panels are cited), condensed (i.e. non coding RNA) or shortened/removed (RNA editing). Other sections need to be better explained (i.e. related to conde present in Dyad). See also other Figure restructuring suggestions (REV#4).

*Reviewer #1 (Recommendations for the authors):*

Overall the manuscript is complex, and may be hard to follow for a general audience, which precludes grasping its overall message. Yet, the work raises some relevant observations that are of interest to colleagues working on fungi as well as other organisms exhibiting complex multicellularity. Particularly, the data appears to downplay the developmental hourglass concept that had been amply proposed in animals and plants, and lately also in fungi. Importantly, in the study by Merényi et al., some of the examined fungi revealed profiles compatible with a developmental hourglass (P. ostreatus, M. kentingensis and A. bisporus), whereas the other datasets argued otherwise, Thus, it will be interesting to see what future studies by this and other groups show regarding the mechanisms incorporating genetic novelty in developmental programs such as complex multicellularity.

Line 42: "Recent research suggests"

Replace by "Diverse studies suggest"

Line 57: "This has impeded a general synthesis on the genetics of CM in fungi"

Try to rephrase, or expand, this phrase.

Figure 2: "(d) percent sequence identity"

Please check legend/figure Y axis

Line 218: "At the same time, they are significantly underrepresented in the three oldest age categories"

The analysis fails to include the behavior of one of the oldest gene groups (4), which has by far the lowest level of ASE expression.

Line 215: "ASE genes showed a strong and significant overrepresentation in the three youngest gene ages".

Examination of the data does not seem to reflect that, as group 20 has a rather low ASE representation. On the other hand, other young groups (17, 18, and 19) appear to better match that description.

Line 229: "The strongest overrepresentation of ASE genes can be observed among developmentally expressed genes that arose in the last common ancestor of Pleurotus (gene age 18)"

The circle size of group 18 is by far the smallest one, indicating a small number of proteins included in such group. Would the authors care to comment?

The paragraph starting in Line 247: "In P. ostreatus we identified 166 NATs (8.1%) that showed at least 2-fold higher expression in all fruiting body stages than in vegetative mycelium, comparatively more than genes (4.8%). Such transcripts may regulate the transition from simple multicellularity in VM to complex multicellularity in FB, one of the most significant transcriptomic reprogramming events in the fungal life cycle (Krizsán et al., 2019). Kim et al. (2018) found a similar proportion (21.3%, 547 of 2,574) of lncRNA that might have a role in sexual development of Fusarium graminearum".

This phrase is somehow confusing: the authors specify 8.1%, and then they comment that 21.3% is a similar proportion.

The paragraph starting in Line 247:

This may be a misleading comparison as they are paralleling their analyses on NATs to lncRNAs (which only in some cases may correspond to antisense RNAs).

Line 249: "comparatively more than genes (4.8%)"

Genes? Probably the authors are trying to imply conventional coding genes.

Line 253: "Nevertheless, only 39.1% of P. ostreatus and 4.1% of Pt. gracilis of the NAT-possessing genes show developmental regulation."

It is not clear what the degree of developmental regulation is: (>4 …>2 ?)

The paragraph starting in Line 268: Please avoid using lncRNAs and NATs as equivalent groups, since not all lncRNAs correspond to NATs. Therefore, it becomes confusing to distinguish when the authors are exclusively taking about NATs and when they are not.

Line 411: "We found that in P. ostreatus ASE is characteristic of young genes and likely arises from promoter divergence, which creates a cellular environment with divergent cis regulatory alleles but identical trans-regulatory elements".

This is an interesting observation and hypothesis. As the authors analyses includes promoter divergence, the discussion (or even the results) would benefit from a couple of clear examples. Although there is rather scarce information of cis-trans pairs specificity in filamentous fungi (even less in basidiomycetes), this can be inferred for closely related DNA Binding Domains (DBDs) (see i.e. 25215497 or 31742324)

Line 440: "Our strategy also identified regulatory genes which are developmentally expressed only in CM species, including certain transcription factors reported in mushrooms (e.g. hom1, fst3, fst4, wc1)"

While these examples are highlighted in the discussion, they are not clearly described in the Results section. This should be emended, particularly as it would help to bring down to earth some of the general results.

*Reviewer #2 (Recommendations for the authors):*

Overall, I found it difficult to find the main message of the paper in the data. The authors carry out extensive bioinformatic analyses of their RNA-Seq data, and go further to categorize different types of transcripts than most work, however while documenting the levels of these different transcript types there did not emerge a clear message about what types of transcripts are linked to multicellularity. There is context that is missing about the genes not just differentially expressed in the stages but those that are known to play a key role in development and how they are regulated.

The message would be more clear if the authors streamlined the manuscript to focus on their main findings. There are a very large number of supplementary table and figure files, and the data presented is not always clearly described. The authors should more finely curate what is included, removing data that is not essential (as it is not described or described only very briefly, such as the lack of clustering of differentially expressed genes, and shorten those that are reporting negative results, such as the section on RNA editing).

There is also more context needed in explaining the stages that were sampled (see comments below) and how regulation differs between stages. From Figure 1, VM appears to be very different from the others, a higher TAI and the low expression of meiotic genes. How can the transcriptional changes be related back to the cellular or multicellular structure of these stages?

There appear to be conflicting data assessing a developmental hourglass, first in Figure S5 (line 112-113) that rejects a general hourglass using the TAI and then support for an hourglass through analysis of gene age (line 294-7, Figure 4b). How can these results be understood together?

Overall, it was difficult to assess the likely importance and certainty of the predictions of the antisense transcripts. The whole section (234-282) describes a thorough analysis of these transcripts, however the low conservation raises questions of whether the majority are transcriptional noise. Without more support for a role for these transcripts, perhaps focusing on those that are the most highly expressed, this section seems preliminary and should be condensed.

There were major issues with data presentation that the authors need to address. First, many figure panels were discussed out of order, leading to confusion when looking through each figure while reading the text. The authors should consider reordering panels and splitting figures so that the flow is smoother. For example, the text refers to Figure 2c, then back to Figure 1, then Figure 2d , Figure 2e, followed by Figure 2b, with Figure 2A never discussed. Second, there are callouts for panels that are missing or incorrect. There is no panel 1d (line 115, perhaps this refers to one of the two panels under 1b?) or 4g (line 319), and the callouts for many of the panels of Figure 4 appear incorrect (for example 4c at lines 295 and 298 appears to refer to 4b, 4c at line 309 appears to refer to 4d). Figure S19 is clipped (parts a and b) in the pdf version provided for review. I recommend adding a main figure panel of the species tree of the transcriptomes compared with inferred CM node transitions highlighted; many analyses in the paper are in the context of the species tree and this should be shown more clearly in a main figure. Figure 3, which has the species tree however too small to view, has a different issue of appearing to make a fairly straightforward point low conservation of ASE with the exception of one species comparison, however the different scale between the two panels does not support direct comparisons to sense transcripts, and it is difficult to understand why the figure is shown if there is not a larger point about types of sense transcripts that are highly conserved. Based on this I would recommend Figure 3 be moved to the supplement.

Line 102: The differential expression analysis needs to be described in full in the methods (line 502 only references an earlier paper). Currently there is no context to understand what comparisons were used to define the developmentally expressed genes. Please also clarify how Pt is "within the range reported" earlier as the 4FC value appears an outlier in Figure S4.

Lines 108-109: This introduction of stages would benefit from more detain as to the major transitions vs growth points sampled.

Line 112: A. ostoye appears to have a pattern very similar to A. bisporus- how were hourglass patterns defined, and were the sampling conditions likely to have contributed to some species not showing this pattern?

Line 113: Figure 1b needs a more detailed legend, possibly split into 2 panels. Where is it described how meiotic genes were selected? How does this show the "highest level of transcriptome conservation"?

Line 142: What are "PC15 and PC9 nuclei"? Please also clarify the genome used as a reference including the source of the data (ie accession).

L159: While the GO terms reported here are found in some versions of the tables/figure, they do not appear to be the most significant results and are lost in some comparisons (ie the tab of only FBDR4 in Table S5). There needs to be more context here of the overall patterns to avoid selective reporting, and also clearer guidance for how to interpret the different results shown in the multiple analyses in Tables S5-6.

Line 216; Explain how gene age is calculated (methods and Figure 2b legend). While three of the youngest appear highly over-represented for ASE, that does not appear to be the case for gene age 20. Are stages 4 and 20 under-represented by based on the species tree (ie v few genes in these sets)?

Line 312-3: There appears to be some circular logic here, as shared orthogroups would by definition pre-date all species. Please clarify.

Line 315-7: These gene categories are difficult to match to the figure.

Line 365: Is C. neoformans the only low complexity species available for comparison in this data set? Some gene losses may more relate to the specialization of these species as human pathogens and notably smaller gene sets compared with many Basidiomycetes.

In the MDS plots (FiguresS3, S7, S20), it is difficult to clearly see how replicates are clustered. Can the authors add a data point for each measurement and offset the names so that they overlap less. It would also be helpful to comment more on the relationships of samples that appear to overlap, ie whether they are from closely related developmental stages or samples.

The authors have deposited the RNA-Seq data in the GEO and also made code available on Dryad. Connecting to Dryad, to see what was there, I was surprised by how much code they have made available as this isn't clearly described in the methods section. It would be helpful for readers to call out more specifically code that is available in Dryad (for example, referring to a "custom awk script" at line 623).

The authors reference using 109 genomes from the JGI genome portal. Are all of these public without restrictions or approved for use in this study? JGI has recently raised concerns with use of some of their data that was under use restrictions.

There are two references that are not correctly formatted as they have no source, Alexa et al. and Bembom et al.

*Reviewer #3 (Recommendations for the authors):*

Figure 1 the highest level of transcriptome conservation coincided with mitotic/meiotic gene upregulation. How would the TAI plots look like after filtering out these mitotic/meiotic genes?

Developmental gene clusters – I found the identification of hotspots not very clear and not very convincing- This section is short and does not seem very useful for the argument of the paper. Could the authors clarify they used this information in subsequent analyses?

L142 36.5% of reads were not able to be assigned to either parental genomes (Table S3) > where do they come from?

Figure 2 Please define EE in figure legend, please indicate here how many genes in the EE, S2 and S4 groups "developmental transcriptomes are dominated by old and young genes in all species, creating 'U' shaped distributions" Could this be due to the strata selected for the phylostratigraphic analysis, that range from species specific to dikarya (and not older) – how many genes are represented in each stratum?

*Reviewer #4 (Recommendations for the authors):*

It would be helpful to explain how the orthology and RBH detection applied is sufficient as compared to other tools for orthology detection, eg OrthoFinder (https://genomebiology.biomedcentral.com/articles/10.1186/s13059-019-1832-y), which appear to be one of the best tools for this approach based on empirical testing and simulations.

The Figure 3 presentation is dense for it shows all the genes and similarity computations in one figure, but it would be helpful to indicate that the similarity comparisons are being made within a species? the labels of species and gene and whether the pairwise comparisons are within a species? I am still left confused as to what is being plotted as I cannot quite understand the NAT interpretation from.

Figure 5 is very dense, it took me a while to understand what was being represented, but I'm not sure how to suggest a simplification. It might be useful to order the species phylogenetically instead of alphabetically along the x-axis? We expect Cryptococcus to behave differently from the rest as it does not have CM.

The Supplementary files appear to represent what were needed to perform the analyses – I did not have a chance to review these in detail at this stage but the only suggestion is also commit these to a code repository like github which can be used to track changes and support more easy integration of these tools into other researchers' workflows.

It isn't clear in Figure S22 if the sense and antisense transcript are shared on the same row as one looks across the two heatmaps? I assume so but there is no way to check since they have different effective transcript IDs?

Some of the y axis labels are trimmed in the supplemental figure heat maps (eg S28)

Figure 29, "number of relative gene ages " in the legend – it is not really clear what this means without better understanding it is a lookup figure for figure 5?

[Editors' note: further revisions were suggested prior to acceptance, as described below.]

Thank you for submitting your article "Gene age predicts the transcriptional landscape of sexual morphogenesis in multicellular fungi" for consideration by *eLife*. Your article has been reviewed by 2 peer reviewers, and the evaluation has been overseen by a Reviewing Editor and Patricia Wittkopp as the Senior Editor. The reviewers have opted to remain anonymous.

Essential revisions:

The new version of the manuscript has included mayor changes in the text, figures and overall narrative of the results, which help to better transmit the main messages of the work. Nevertheless, there are still some aspects that need additional modification to avoid overstatements, as some of the analyses lead to interesting speculations which are not necessarily bulletproof conclusions. To turn the former in the latter one would need experimental validation. Nevertheless, that would be a daunting task considering the characteristics of the fungi under study. The alternative is to tone down what may be presented as conclusions and present them more as speculations (i.e. the datasets strongly suggest… the analyses allow us to speculate that…).

1. For example, reviewer 3 indicates: "The identification of CM-related genes does not include experimental validation (either partial or indirect). The proposed list is therefore speculative, and any evolutionary inference is dependent on the validity of this CM-related gene candidates. I acknowledge validation can be seen as follow up work, in which case I would urge the authors not to overstate the validity of their candidate list'. Related to that, Rev#2 also indicates the lack of in depth-analysis of the relevant proposed genes.

2. Likewise, some aspects of the text may need to be better connected/ presented. Indeed, even though the datasets on global expression are extremely interesting a main point of the manuscript on how gene age, NATs and ASE contribute to CM still needs to be better explained.

3. Along the same lines, it is important not to generalize the impact of the fungi as they may or may not be applicable to CM in all fungi (basidiomycetes, etc).

Regarding the data on PWM, the authors may be correct that there is too much noise in the current data to consider including it at some point. Nevertheless, such analyses, complemented with reporter systems and evaluating the different alleles' promoters may be an interesting future study.

*Reviewer #2 (Recommendations for the authors):*

The authors have addressed the points raised in the prior review and made major updates to the manuscript that have improved the clarity of the main findings. In my view, the only area not addressed in depth concerns the summary comment 2, recommending that " an in-depth analysis of the most relevant genes and pathways is necessary". The authors have added some additional detail on gene functions and homologs, however this seems limited, and the authors note that this is in part due to the lack of prior predictive functional information for such developmental genes.

*Reviewer #3 (Recommendations for the authors):*

In this revised manuscript, Merenyi et al. have provided very useful new elements of context and logic to clarify the rationale of their study. I appreciated very much making the list of CM gene candidates explicit and the comments on a number of molecular functions that may be relevant. There has been an effort to connect pieces of the study together that allowed me to get a better view of the overall reasoning and implications of the work. Nevertheless, I believe there is a need to further clarify the overall study objectives and logic, and to avoid overstatements.

Indeed, I still find the study objectives fuzzy and the logic somewhat confusing. L77-79 states "The goal of this study was to systematically tease apart the components and driving forces of transcriptome evolution in CM fungi and to identify conserved CM-related genes in the Agaricomycetes", L60-66: "how species-specific and conserved genes, long non-coding RNA, natural antisense transcripts, allele specific expression, small RNA, […] contribute to CM is not known". In the response to reviewers' comments: "one of the novel aspects of our manuscript is the identification of a very specific set of candidate genes". In my view these are incompletely met in this manuscript for the following reasons.

1. The identification of CM-related genes does not include experimental validation (either partial or indirect). The proposed list is therefore speculative, and any evolutionary inference is dependent on the validity of this CM-related gene candidates. I acknowledge validation can be seen as follow up work, in which case I would urge the authors not to overstate the validity of their candidate list.

2. The distinction between CM gene candidates and sexual development genes comes at the end of the Results section with the comparison with C. neoformans, after evolutionary/conservation analyses are presented: While CM-gene candidates are identified in the section L524 and on ("To distinguish genes related to basic sexual processes […] from those related to CM"), some analyses are presumably targeted to CM gene candidates earlier in the manuscript, such as L166 ("To examine how these young genes contribute to the CM transcriptome"), L194 ("young genes […] do not have a uniform contribution to CM development"), L288 ("These examples highlighted the importance of allele specific expression in generating expression variance in CM-related genes"). With an aim to study evolutionary/conservation patterns in CM genes, I would have expected the identification of CM genes to be reported first.

From the response to previous reviewers' comments, I understand that the TAI pattern and NAT occurrence are not significantly different between CM gene candidates and sexual development genes, while ASE is characteristic for young genes regardless of their putative involvement in CM. Although global patterns are interesting, how gene age, NATs and ASE contribute to CM therefore remains elusive to me.

3. In a number of places, the authors tend to generalize their findings to multiple fungal lineages while some of the analysis only include one or two species (P. ostreatus and Pt. gracilis). For instance, L77 "the components […] of transcriptome evolution in CM fungi" such as NATs and ASE are studied in P. ostreatus and Pt. gracilis and generalization to CM fungi might be premature. L22-25 "We here reveal that allele-specific expression, natural antisense transcripts […] act in concert to shape the transcriptome of complex multicellular fruiting bodies of fungi" is in fact shown for P. ostreatus only. In the title "the transcriptional landscape of sexual morphogenesis" has been studied mostly in P. ostratus and may not generalize to "multicellular fungi". Given the diverse patterns revealed by the TAI analysis encompassing multiple species, whether the ASE and RNA editing patterns are "general principles" (L213) remains speculative.

4. The "predictive value" of the study put forward in the title and abstract is confusing to me. e.g. L25 "evolutionary age predicts, to a large extent, the behaviour of a gene in the CM transcriptome", it is not completely clear to me what can be predicted. I am rather used to see the term "predict" used in relation to a formal model the performances of which are quantitatively evaluated. Generally speaking, the connection between the manuscript title, the proposed study objectives and major findings from the work is quite cryptic to me.

---

## [Author Response]

Essential revisions:1) The manuscript needs to better integrate and streamline the analyses to clarify the paper's main question. For example, as indicated by REV#2, a clear message about what types of transcripts are linked to multicellularity is missing: instead of just mentioning DEGs in critical stages, more context is needed regarding the ones playing a key role in development and how they are regulated.

We reworked the manuscript to highlight the main messages better and in the revised version provide more context on multicellularity-related transcripts in the sections on ASE and sexual and CM development (see lines 272-288, 615-626, 631-638 and Figure 5, Figure 8 figure supplement 1). We highlighted development-related transcription factors, hydrophobins and other genes associated with multicellularity in fungi. We also note that the genetic bases of multicellular development are very poorly known in the Basidiomycota, as a consequence, the vast majority of genes in conserved orthogroups are new predictions without functional verification at the moment. Therefore, we interpret these results conservatively in the manuscript.

2) An in-depth analysis of the most relevant genes and pathways is necessary. While one could think of experimental validation (i.e. knocking out/ overexpressing candidate genes), taking in consideration that the paper has a main evolutionary context, a more in-depth analysis of genes/regulatory motifs/pathways and a better description of the associated gene programs could be an alternative. Likewise, they need to better elaborate some connections between sections of the manuscript (i.e. hourglass model and ASE), and provide case examples for other key aspects of the manuscript (i.e point 4 by REV#3)

We reworked the text to better connect parts of the manuscript and highlight logical connections better. We also placed more emphasis on the in-depth analysis of genes related to complex multicellularity (see lines 615-626, 631-638), including case examples. We note, however, that current functional characterizations (e.g. GO/KEGG systems) of fungal genes hardly extend to developmental/complexity related aspects. Therefore, in-depth analyses of multicellularity-related genes are possible only for a quite limited set of genes, whereas inclusive and systematic analyses are impossible at the moment, because key developmental pathways are not known (unlike in the Ascomycota). This is an inherent challenge for mushroom developmental biology, and we hope that our current work contributes to closing this gap in knowledge.

3) The reviewers suggested that some of the data should be removed from the manuscript as it does not clearly contribute to the main point that is trying to make. This also applies to supplementary data. Moreover, they indicated that there is a very large number of supplementary files (tables and figures), where the presented data is not always clearly described. The authors should more finely curate what is included, removing data that is not essential.

We agree that the manuscript was trying to achieve too much and in the revised version we removed several sections (e.g. developmental gene clusters -> Appendix 1; Natural antisense transcripts -> Appendix 2 and RNA editing -> Appendix 3) and reconsidered the content of the supplementary data carefully (merged Figure S9-10, Table S5-6, Table S8-9, Table S10-11, removed: Figure S8, S11 moved to Dryad Figure S29 Table S1-4, S7, condensed: Figure S28, Figure S26). Also, we restructured big figure panels into primary and “child” figures following *eLife* presentation logic.

4) Several of the sections need to be restructured (paying attention on how/when Figure panels are cited), condensed (i.e. non coding RNA) or shortened/removed (RNA editing). Other sections need to be better explained (i.e. related to conde present in Dyad). See also other Figure restructuring suggestions (REV#4).

We restructured the manuscript significantly and reconsidered all figures and supplementary files in accordance with the Reviewers’ suggestions. We feel that the manuscript has improved significantly.

Reviewer #1 (Recommendations for the authors):Overall the manuscript is complex, and may be hard to follow for a general audience, which precludes grasping its overall message. Yet, the work raises some relevant observations that are of interest to colleagues working on fungi as well as other organisms exhibiting complex multicellularity. Particularly, the data appears to downplay the developmental hourglass concept that had been amply proposed in animals and plants, and lately also in fungi. Importantly, in the study by Merényi et al., some of the examined fungi revealed profiles compatible with a developmental hourglass (P. ostreatus, M. kentingensis and A. bisporus), whereas the other datasets argued otherwise, Thus, it will be interesting to see what future studies by this and other groups show regarding the mechanisms incorporating genetic novelty in developmental programs such as complex multicellularity.Line 42: "Recent research suggests".Replace by "Diverse studies suggest".

Done.

Line 57: “This has impeded a general synthesis on the genetics of CM in fungi”.Try to rephrase, or expand, this phrase.

Done.

Figure 2: "(d) percent sequence identity".Please check legend/figure Y axis.

Y axis corrected.

Line 218: “At the same time, they are significantly underrepresented in the three oldest age categories”.The analysis fails to include the behavior of one of the oldest gene groups (4), which has by far the lowest level of ASE expression.

We appreciate this comment, we included age group 4 (which shows the strongest underrepresentation) in the discussion now.

Line 215: "ASE genes showed a strong and significant overrepresentation in the three youngest gene ages".Examination of the data does not seem to reflect that, as group 20 has a rather low ASE representation. On the other hand, other young groups (17, 18, and 19) appear to better match that description.

We updated this part. After rerunning the analysis, we find that in fact, age group 20 also aligns with the trendline. The difference comes from a simple change in the analysis method: in our previous analysis, we treated all genes that did not show clear allele-specific expression (based on AS<0.31 or AS>0.68 L1010-1014) as equally expressed (EE), whereas in the new analysis we only consider a gene EE if its expression high enough to decide that the expression between alleles is significantly non-different (0.31 < AS <0.68). This way, although overall we analyzed fewer genes (some genes were lost because they became neither significantly EE nor ASE), but the analysis is conceptually more sound and the patterns are clearer. The explanation why gene age 20 showed the highest change, is that gene age 20 contained the most of those genes that we excluded because of the minimal expression requirement. We updated the text accordingly.

Line 229: "The strongest overrepresentation of ASE genes can be observed among developmentally expressed genes that arose in the last common ancestor of Pleurotus (gene age 18)".The circle size of group 18 is by far the smallest one, indicating a small number of proteins included in such group. Would the authors care to comment?

Thank you for this remark, we rewrote this sentence.

The paragraph starting in Line 247: “In P. ostreatus we identified 166 NATs (8.1%) that showed at least 2-fold higher expression in all fruiting body stages than in vegetative mycelium, comparatively more than genes (4.8%). Such transcripts may regulate the transition from simple multicellularity in VM to complex multicellularity in FB, one of the most significant transcriptomic reprogramming events in the fungal life cycle (Krizsán et al., 2019). Kim et al. (2018) found a similar proportion (21.3%, 547 of 2,574) of lncRNA that might have a role in sexual development of Fusarium graminearum”.This phrase is somehow confusing: the authors specify 8.1%, and then they comment that 21.3% is a similar proportion.

In this case, 8,1% refers to only the mycelium to primordium transition, I 57,4 and 16,5% (Appendix 2 line 61-63) which are relevant. Nevertheless, even these are not very similar, so we rephrased this section.

The paragraph starting in Line 247:This may be a misleading comparison as they are paralleling their analyses on NATs to lncRNAs (which only in some cases may correspond to antisense RNAs).

We agree and added a note on this to the sentence.

Line 249: "comparatively more than genes”(4.8%)" Genes? Probably the authors are trying to imply conventional coding genes.

Yes, corrected, thank you.

Line 253: "Nevertheless, only 39.1% of P. ostreatus and 4.1% of Pt. gracilis of the NAT-possessing genes show developmental regulation."It is not clear what the degree of developmental regulation is: (>4 …>2 ?)

We meant at fold change >4, and added a note on this to the text.

The paragraph starting in Line 268: Please avoid using lncRNAs and NATs as equivalent groups, since not all lncRNAs correspond to NATs. Therefore, it becomes confusing to distinguish when the authors are exclusively taking about NATs and when they are not.

Thank you, we rephrased it.

Line 411: "We found that in P. ostreatus ASE is characteristic of young genes and likely arises from promoter divergence, which creates a cellular environment with divergent cis regulatory alleles but identical trans-regulatory elements".This is an interesting observation and hypothesis. As the authors analyses includes promoter divergence, the discussion (or even the results) would benefit from a couple of clear examples. Although there is rather scarce information of cis-trans pairs specificity in filamentous fungi (even less in basidiomycetes), this can be inferred for closely related DNA Binding Domains (DBDs) (see i.e. 25215497 or 31742324)

We appreciate the articles and ideas. The detailed answers were placed in the public review part.

Line 440: "Our strategy also identified regulatory genes which are developmentally expressed only in CM species, including certain transcription factors reported in mushrooms (e.g. hom1, fst3, fst, wc1)".While these examples are highlighted in the discussion, they are not clearly described in the Results section. This should be emended, particularly as it would help to bring down to earth some of the general results.

We added information on key TF genes to the Results section (lines 1170-1176) and Table S2.

Reviewer #2 (Recommendations for the authors):Overall, I found it difficult to find the main message of the paper in the data. The authors carry out extensive bioinformatic analyses of their RNA-Seq data, and go further to categorize different types of transcripts than most work, however while documenting the levels of these different transcript types there did not emerge a clear message about what types of transcripts are linked to multicellularity. There is context that is missing about the genes not just differentially expressed in the stages but those that are known to play a key role in development and how they are regulated.The message would be more clear if the authors streamlined the manuscript to focus on their main findings. There are a very large number of supplementary table and figure files, and the data presented is not always clearly described. The authors should more finely curate what is included, removing data that is not essential (as it is not described or described only very briefly, such as the lack of clustering of differentially expressed genes, and shorten those that are reporting negative results, such as the section on RNA editing).

We appreciate the Reviewer’s suggestions and have made structural changes to the manuscript to improve the storyline. The main message of the paper is that gene age determines, to a large extent, transcriptomic patterns (first half of ms, NATs, ASE, gene age distributions) and that correcting for this phenomenon helped identifying a narrowed set of genes that might be responsible for sculpting complex multicellular fruiting bodies. In the revised version we highlight these broad conclusions even more. We shortened parts that are circumstantial to the main story and reorganized supplemental data. We also provide more context on the link between transcripts/genes and multicellularity, although we note that this is very poorly known in fungi and one of the novel aspects of our manuscript is the identification of a very specific set of candidate genes.

There is also more context needed in explaining the stages that were sampled (see comments below) and how regulation differs between stages. From Figure 1, VM appears to be very different from the others, a higher TAI and the low expression of meiotic genes. How can the transcriptional changes be related back to the cellular or multicellular structure of these stages?

Since the specific stages we use are only interesting to mycologists, we prefer to explain the sampled stages in more detail in the Methods section, where we added more details on the sampled stages (see lines 1481-1488). Due to the shortage of information on complex multicellularity in fungi and in general in functional annotations of genes outside the major model organisms (mostly *S. cerevisiae*, Sch. pombe), it is impossible to speculate on broad differences in regulation between VM and fruiting bodies, or on what gene functions might contribute to the high TAI values in vegetative mycelia. Vegetative mycelia is composed mostly of non-dividing cells and is focused on nutrient acquisition and translocation towards the fruiting body, whereas in fruiting bodies active cell proliferation and differentiation takes place. Considering that the emergence of fruiting bodies is a major transcripitional reprogramming event, it is possible that likewise genome-wide regulatory changes (e.g. in chromatin-regulation or -accessibility, changes in histone code) also happen, however, we currently don't have any information on that. Vonk and Ohm (https://pubmed.ncbi.nlm.nih.gov/33854169/) recently provided an example on histone 3 lysine 4 methylation, which showed that dikaryon formation is indeed associated with changes to histone methylation. The field would need more of this kind of study (e.g. with ATAC-Seq, DNAse-Seq), before we can confidently discuss the modes of gene expression regulation in different stages.

There appear to be conflicting data assessing a developmental hourglass, first in Figure S5 (line 112-113) that rejects a general hourglass using the TAI and then support for an hourglass through analysis of gene age (line 294-7, Figure 4b). How can these results be understood together?

This is a misunderstanding by the Reviewer. On line 294-297 we refer to the age distribution of genes in the genome, which is not an assessment of the developmental hourglass. In the case of developmental hourglass, the ‘waist’ is caused by young genes being expressed in early and late but not intermediate phases of development. The paragraph on lines 294-7 (and corresponding previous figure S26, now Figure 2) talks about gene ages, where the ‘waist’ is caused by the scarcity of genes in intermediate gene age categories. Every organism has a lot of ancient and a lot of young genes, which is a characteristic of nearly all species. The reviewer is right in that these gene ages are indeed used in the TAI calculations for the developmental hourglass, but are not equal to those. We made a note on this in the manuscript.

Overall, it was difficult to assess the likely importance and certainty of the predictions of the antisense transcripts. The whole section (234-282) describes a thorough analysis of these transcripts, however the low conservation raises questions of whether the majority are transcriptional noise. Without more support for a role for these transcripts, perhaps focusing on those that are the most highly expressed, this section seems preliminary and should be condensed.

We agree with the reviewer that the antisense transcription is hard to link to multicellularity. Therefore, we moved this section to the supplement. The observation that they show dynamic expression patterns provides some evidence, although it is true, as we pointed out in the manuscript, that a fraction of them may also be transcriptional noise. Based on the structure, length, and expression profile of the detected NATs (see Appendix 2), however, we can exclude the possibility that they are all transcriptional noise, probably several of them are involved in regulating multicellular events. To check this, we reanalyzed the expression dynamics of NATs under a stricter parameter set, essentially excluding all samples with <20 FPKM (instead of 4 in the original analysis). This exercise should work against transcripts that arise from transcriptional noise. We found that the proportions changed, but developmentally regulated NATs were not completely eliminated: 31.9% of NATs are developmentally regulated (57.4% when FPKM<4 is the cutoff). Our results are supported by literature data where there is clear evidence for a developmental role of certain NATs (and lncRNA), although in all cases, these are only based on individual case studies.

There were major issues with data presentation that the authors need to address. First, many figure panels were discussed out of order, leading to confusion when looking through each figure while reading the text. The authors should consider reordering panels and splitting figures so that the flow is smoother. For example, the text refers to Figure 2c, then back to Figure 1, then Figure 2d, Figure 2e, followed by Figure 2b, with Figure 2A never discussed. Second, there are callouts for panels that are missing or incorrect. There is no panel 1d (line 115, perhaps this refers to one of the two panels under 1b?) or 4g (line 319), and the callouts for many of the panels of Figure 4 appear incorrect (for example 4c at lines 295 and 298 appears to refer to 4b, 4c at line 309 appears to refer to 4d). Figure S19 is clipped (parts a and b in the pdf version provided for review).

We have corrected these mistakes, thanks for noting them.

I recommend adding a main figure panel of the species tree of the transcriptomes compared with inferred CM node transitions highlighted; many analyses in the paper are in the context of the species tree and this should be shown more clearly in a main figure. Figure 3, which has the species tree however too small to view, has a different issue of appearing to make a fairly straightforward point low conservation of ASE with the exception of one species comparison, however the different scale between the two panels does not support direct comparisons to sense transcripts, and it is difficult to understand why the figure is shown if there is not a larger point about types of sense transcripts that are highly conserved. Based on this I would recommend Figure 3 be moved to the supplement.

Thank you for the suggestions. We added a species tree to Figure 1 where we marked the emergence of complex multicellularity according to Merenyi et al. 2020 (https://pubmed.ncbi.nlm.nih.gov/32191325/ ) and an example for gene age numbering. Figure 3 was moved to the Appendix 3

Line 102: The differential expression analysis needs to be described in full in the methods (line 502 only references an earlier paper). Currently there is no context to understand what comparisons were used to define the developmentally expressed genes. Please also clarify how Pt is "within the range reported" earlier as the 4FC value appears an outlier in Figure S4.

Thank you, we corrected it. We rephrased the text relevant to Pt. gracilis.

Lines 108-109: This introduction of stages would benefit from more detain as to the major transitions vs growth points sampled.

We added more detail to this chapter.

Line 112: A. ostoye appears to have a pattern very similar to A. bisporus- how were hourglass patterns defined, and were the sampling conditions likely to have contributed to some species not showing this pattern?

We considered a TAI profile as an hourglass pattern if both early (VM) and late (FB) stages had higher TAI values than middle stages. Our sampling covers the entire developmental trajectory of each species, so we think that if there was a high conservation time point during development we would have caught it using our samples. Although some species have fewer time points (e.g. Pt. gracilis), this reflects the simplicity of their development, so we think our sampling is complete in these cases too. In accordance with this, to our best knowledge, sampling conditions have not yet been described in the literature to interfere with the detection of hourglass patterns. Therefore, we are confident that species in which we did not detect an hourglass-shaped TAI profile indeed do not show this, should a different sampling regime be applied.

We note that in the revised manuscript we transitioned from a phylostratum-based TAI calculation (i.e. phylostrata numbered based on node numbers) to a sequential numbering of phylostrata (1,2,3…20), which caused slight differences in the TAI profiles, but the overall message did not change.

Line 113: Figure 1b needs a more detailed legend, possibly split into 2 panels. Where is it described how meiotic genes were selected? How does this show the "highest level of transcriptome conservation"?

We redesigned Figure 1 completely. Meiotic genes were selected based on Burns et al. 2010 and by orthology to meiotic genes described in *S. cerevisiae* (Cherry et al. 2012). After a more in-depth analysis of drivers of the hourglass pattern, we concluded that in fact mitotic/meiotic genes contributed less than we expected, so we removed this part from the ms.

Line 142: What are "PC15 and PC9 nuclei"? Please also clarify the genome used as a reference including the source of the data (ie accession).

Thank you, we corrected it. PC15 and PC9 refer to the two haploid (monokaryon) parent strains whose mating yields the fruiting dikaryon.

L159: While the GO terms reported here are found in some versions of the tables/figure, they do not appear to be the most significant results and are lost in some comparisons (ie the tab of only FBDR4 in Table S5). There needs to be more context here of the overall patterns to avoid selective reporting, and also clearer guidance for how to interpret the different results shown in the multiple analyses in Tables S5-6.

We, rephrased interpretation of enriched GO terms and merged Table S5 and S6 (now Table S1).

Line 216; Explain how gene age is calculated (methods and Figure 2b legend). While three of the youngest appear highly over-represented for ASE, that does not appear to be the case for gene age 20. Are stages 4 and 20 under-represented by based on the species tree (ie v few genes in these sets)?

Our gene age definition follows standard phylostratigraphic methods (see Domazet-Loso et al. 2007 https://pubmed.ncbi.nlm.nih.gov/18029048/), to clarify this, we put short descriptions in both the Methods and in the legend of Figures 1,2, and 7. We updated the Mann-Kendall analysis part of ms. After rerunning the analysis, we find that in fact, age group 20 also aligns with the trendline. The difference comes from a simple change in the analysis method: in our previous analysis, we treated all genes that did not show clear allele-specific expression (based on AS<0.31 or AS>0.68 L234) as equally expressed (EE), whereas in the new analysis we only consider a gene EE if its expression high enough to decide that the expression between alleles is significantly non-different from equal (0.31 < AS <0.68). This way, although overall we analyzed fewer genes (some genes were lost because they became neither significantly EE nor ASE), but the analysis is conceptually more sound and the inferred patterns are nicer. The explanation why gene age 20 showed the highest change, is that gene age 20 contained the most of those genes that we excluded because of the minimal expression requirement. We updated the text accordingly.

Line 312-3: There appears to be some circular logic here, as shared orthogroups would by definition pre-date all species. Please clarify.

We deleted the sentence.

Line 315-7: These gene categories are difficult to match to the figure.

In the revised version we refer to Figure 8/c-d. Also, we created Table S2/b-c, which contains significantly overrepresented GO terms for clarify this issue.

Line 365: Is C. neoformans the only low complexity species available for comparison in this data set? Some gene losses may more relate to the specialization of these species as human pathogens and notably smaller gene sets compared with many Basidiomycetes.

Yes, C. neoformans is the only low complexity species where a whole and detailed RNAseq dataset is available. The Reviewer is right that the absence of a gene from C. neoformans could be because of gene loss in this species or later origin in the Agaricomycetes. We noted this on line 533-535.

In the MDS plots (FiguresS3, S7, S20), it is difficult to clearly see how replicates are clustered. Can the authors add a data point for each measurement and offset the names so that they overlap less. It would also be helpful to comment more on the relationships of samples that appear to overlap, i.e. whether they are from closely related developmental stages or samples.

We corrected the plots, and added more explanation in the main text.

The authors have deposited the RNA-Seq data in the GEO and also made code available on Dryad. Connecting to Dryad, to see what was there, I was surprised by how much code they have made available as this isn't clearly described in the methods section. It would be helpful for readers to call out more specifically code that is available in Dryad (for example, referring to a "custom awk script" at line 623).

All scripts which we had uploaded to Dryad are now mentioned in the Materials and methods or figures (like Figure 10). We also moved supplementary tables and wrote a clearer explanation for scripts uploaded to Dryad.

The authors reference using 109 genomes from the JGI genome portal. Are all of these public without restrictions or approved for use in this study? JGI has recently raised concerns with use of some of their data that was under use restrictions.

All but one of the 109 genomes are published, we added the publications with pubmed links to Table S3. Aphanobasidium pseudotsugae is unpublished, but we obtained permission from Otto Miettinen (genome PI).

There are two references that are not correctly formatted as they have no source, Alexa et al. and Bembom et al.

Thank you, we corrected it.

Reviewer #3 (Recommendations for the authors):Figure 1 the highest level of transcriptome conservation coincided with mitotic/meiotic gene upregulation. How would the TAI plots look like after filtering out these mitotic/meiotic genes?

To answer this intriguing question, we attempted to decipher if mitotic/meiotic genes indeed drove the hourglass patterns. We find that TAI plots without mitotic/meiotic genes retained the hourglass pattern for P. ostreatus (Author response image 1). After several attempts to identify a well-delimited gene group that drives the hourglass pattern (using filters for specifically U-shaped genes, interacting L-shaped and reverse L-shaped genes, removing oldest, youngest genes, specific functional categories, etc.) we came to the conclusion that we cannot pinpoint a single gene group that drives this pattern. Rather, its appears that signal for the hourglass-shape is diffuse among all genes of P. ostreatus. We note here also that, because P. ostreatus is an exception rather than a rule among the 9 examined Agaricomycetes, we do not devote too much space to deciphering driving forces of its hourglass pattern. Rather, we interpret the overall results that simply because fungal CM development is so different from that of animals and plants, the hourglass (or the early-conservation) model does not apply for fungi.

**Author response image 1. sa2fig1:** Transcriptome Age Index for only meiotic genes and without them of *P. ostreatus* separately.

Developmental gene clusters – I found the identification of hotspots not very clear and not very convincing- This section is short and does not seem very useful for the argument of the paper. Could the authors clarify they used this information in subsequent analyses?L142 36.5% of reads were not able to be assigned to either parental genomes (Table S3) > where do they come from?

All reads were classified into three categories: come from reference PC15 (A) come from reference PC9 (B) or indecisive (C). Indecisive reads still come from P. ostreatus, but do not have enough information (SNPs relative to parental genomes) to be unambiguously classified. The definition of indecisive reads can be found in the Materials and methods section (line 995-998): “We assigned a read as indecisive if (i) Hd>1 from both reference genomes (ii) Hd>15 from any of the reference genomes (too divergent read) or, (iii) if the Hd was equal to both parental genomes.” For further filtering, if too many indecisive reads were detected in a gene (>80%), we considered it as an equally expressed gene (L665). With this strict classification of reads, a significant amount of reads was lost, in order to reach a more reliable dataset.

Figure 2 Please define EE in figure legend, please indicate here how many genes in the EE, S2 and S4 groups "developmental transcriptomes are dominated by old and young genes in all species, creating 'U' shaped distributions" Could this be due to the strata selected for the phylostratigraphic analysis, that range from species specific to dikarya (and not older) – how many genes are represented in each stratum?

Phylostratigraphic studies frequently see a dominance of very young and old genes in any organism. Therefore, we do not expect that the shape of the distribution would have changed if we analyzed more distant species. In a previous paper we used more pre-dikarya species (Krizsan et al. 2019; Figure S28), while Cheng et al. 2015 expressed gene ages across all living organisms (Figure 2a https://academic.oup.com/mbe/article/32/6/1556/1074359#74396897); both studies obtained a similar U-shaped distribution. Including more ancient groups (basal fungi or non-fungal opisthokonts) would have yielded a more resolved pattern about the gene ages, however that would have come also with a cost in computational efficiency and precision. The number of genes in each phylostratum is given in Author response image 2.

**Author response image 2. sa2fig2:** The number of genes in each phylostratum.

Reviewer #4 (Recommendations for the authors):It would be helpful to explain how the orthology and RBH detection applied is sufficient as compared to other tools for orthology detection, eg OrthoFinder (https://genomebiology.biomedcentral.com/articles/10.1186/s13059-019-1832-y), which appear to be one of the best tools for this approach based on empirical testing and simulations.

Indeed, OrthoFinder is one of if not the best in the field of orthology detection, but for our purposes here we needed 1:1 orthologs. OrthoFinder includes (in)paralogs in orthogroups, which would have made the strict 1:1 analyses in this manuscript impossible. We ran OrthoFinder on this dataset and could detect only 2009 single copy orthologue groups, indicating that it’s not an optimal tool for the specific questions addressed in this manuscript.

The Figure 3 presentation is dense for it shows all the genes and similarity computations in one figure, but it would be helpful to indicate that the similarity comparisons are being made within a species? the labels of species and gene and whether the pairwise comparisons are within a species? I am still left confused as to what is being plotted as I cannot quite understand the NAT interpretation from.

We clarified this in the figure legend and moved this figure to the appendix 1.

Figure 5 is very dense, it took me a while to understand what was being represented, but I'm not sure how to suggest a simplification. It might be useful to order the species phylogenetically instead of alphabetically along the x-axis? We expect Cryptococcus to behave differently from the rest as it does not have CM.

We appreciate this comment, We ordered the x axis phylogenetically. Yes, we indeed failed to detect any enrichment of CM-specific genes in Cryptococcus, which confirmed that most of CM-specific genes duplicated or “born” after the split of Tremellomycetes (Cryptococcus)

The Supplementary files appear to represent what were needed to perform the analyses – I did not have a chance to review these in detail at this stage but the only suggestion is also commit these to a code repository like github which can be used to track changes and support more easy integration of these tools into other researchers' workflows.

We moved some supplementary tables to Dryad. All scripts which we uploaded to Dryad were incorporated into the main text (Materials and methods) or supplementary figures (like Figure S12). We also uploaded additional scripts, tables and a clearer explanation of scripts to Dryad.

It isn't clear in Figure S22 if the sense and antisense transcript are shared on the same row as one looks across the two heatmaps? I assume so but there is no way to check since they have different effective transcript IDs?Some of the y axis labels are trimmed in the supplemental figure heat maps (eg S28)

We added clarification to the legend of Figure S22.

Figure 29, "number of relative gene ages " in the legend – it is not really clear what this means without better understanding it is a lookup figure for figure 5?

Yes, this figure provides a lookup of gene age labels used on Figure 5. We rephrased the legend and moved the figure to Dryad (Dryad: Figure D1 https://doi.org/10.5061/dryad.5qfttdz5m).

References

Burns C, Stajich JE, Rechtsteiner A, Casselton L, Hanlon SE, Wilke SK, Savytskyy OP, Gathman AC, Lilly WW, Lieb JD, Zolan ME, Pukkila PJ. 2010. Analysis of the Basidiomycete Coprinopsis cinerea Reveals Conservation of the Core Meiotic Expression Program over Half a Billion Years of Evolution. PLOS Genet 6:e1001135. doi:10.1371/JOURNAL.PGEN.1001135

Cheng X, Hui JHL, Lee YY, Wan Law PT, Kwan HS. 2015. A “developmental hourglass” in fungi. Mol Biol Evol 32:1556–1566. doi:10.1093/molbev/msv047

Cherry JM, Hong EL, Amundsen C, Balakrishnan R, Binkley G, Chan ET, Christie KR, Costanzo MC, Dwight SS, Engel SR, Fisk DG, Hirschman JE, Hitz BC, Karra K, Krieger CJ, Miyasato SR, Nash RS, Park J, Skrzypek MS, Simison M, Weng S, Wong ED (2012) Saccharomyces Genome Database: the genomics resource of budding yeast. Nucleic Acids Res. Jan;40(Database issue):D700-5. [PMID: 22110037]

Kim W, Miguel-Rojas C, Wang J, Townsend JP, Trail F. 2018. Developmental dynamics of long noncoding RNA expression during sexual fruiting body formation in Fusarium graminearum. MBio 9:1–17. doi:10.1128/mBio.01292-18

Krizsán K, Almási É, Merényi Z, Sahu N, Virágh M, Kószó T, Mondo S, Kiss B, Bálint B, Kües U, Barry K, Cseklye J, Hegedüs B, Henrissat B, Johnson J, Lipzen A, Ohm RA, Nagy I, Pangilinan J, Yan J, Xiong Y, Grigoriev I V., Hibbett DS, Nagy LG. 2019. Transcriptomic atlas of mushroom development reveals conserved genes behind complex multicellularity in fungi. Proc Natl Acad Sci U S A 116:7409–7418. doi:10.1073/pnas.1817822116

Lee Y, Ulzurrun GV De, Schwarz EM, Stajich JE. 2020. Genome sequence of the oyster mushroom Pleurotus ostreatus strain PC9 Corresponding authors Yen-Ping Hsueh Institute of Molecular Biology Academia Sinica 128 Academia Road , Section 2 , Nangang Taipei , 115 Taiwan

Merényi Z, Prasanna AN, Wang Z, Kovács K, Hegedüs B, Bálint B, Papp B, Townsend JP, Nagy LG. 2020. Unmatched Level of Molecular Convergence among Deeply Divergent Complex Multicellular Fungi. Mol Biol Evol 37:2228–2240. doi:10.1093/molbev/msaa077

Ohm RA, de Jong JF, de Bekker C, Wösten HAB, Lugones LG. 2011. Transcription factor genes of Schizophyllum commune involved in regulation of mushroom formation. Mol Microbiol 81:1433–1445. doi:10.1111/j.1365-2958.2011.07776.x

Piasecka, B., Lichocki, P., Moretti, S., Bergmann, S., & Robinson-Rechavi, M. (2013). The hourglass and the early conservation models—co-existing patterns of developmental constraints in vertebrates. PLoS genetics, 9(4), e1003476.

[Editors' note: further revisions were suggested prior to acceptance, as described below.]

Essential revisions:The new version of the manuscript has included mayor changes in the text, figures and overall narrative of the results, which help to better transmit the main messages of the work. Nevertheless, there are still some aspects that need additional modification to avoid overstatements, as some of the analyses lead to interesting speculations which are not necessarily bulletproof conclusions. To turn the former in the latter one would need experimental validation. Nevertheless, that would be a daunting task considering the characteristics of the fungi under study. The alternative is to tone down what may be presented as conclusions and present them more as speculations (i.e. the datasets strongly suggest… the analyses allow us to speculate that…).

We appreciate the suggestions of the Reviewers and the Editors and have revised the manuscript accordingly.

1. For example, reviewer 3 indicates: "The identification of CM-related genes does not include experimental validation (either partial or indirect). The proposed list is therefore speculative, and any evolutionary inference is dependent on the validity of this CM-related gene candidates. I acknowledge validation can be seen as follow up work, in which case I would urge the authors not to overstate the validity of their candidate list'. Related to that, Rev#2 also indicates the lack of in depth-analysis of the relevant proposed genes.

We toned down strong statements on the relationship between the identified genes and CM. Our use of the term ‘CM-related’ was unjustified in many cases. We changed the phrasing to reflect the fact that these are functional hypotheses based on expression patterns.

We also added a validation analysis of developmentally expressed genes. Although this might not be the in-depth analysis that Rev#2 envisioned, it shows that our bioinformatic pipelines identify genes relevant to CM and the currently available toolbox for Basidiomycetes does not allow us to perform a more thorough functional analysis without sacrificing analytical stringency.

2. Likewise, some aspects of the text may need to be better connected/ presented. Indeed, even though the datasets on global expression are extremely interesting a main point of the manuscript on how gene age, NATs and ASE contribute to CM still needs to be better explained.

We added clarification on this to the ms; we believe the parts are better connected now and explain the relationships between gene age, ASE and developmental expression.

3. Along the same lines, it is important not to generalize the impact of the fungi as they may or may not be applicable to CM in all fungi (basidiomycetes, etc).

We removed unjustified generalisations from the ms.

Regarding the data on PWM, the authors may be correct that there is too much noise in the current data to consider including it at some point. Nevertheless, such analyses, complemented with reporter systems and evaluating the different alleles' promoters may be an interesting future study.

We agree and this is a research direction we are already moving towards.

Reviewer #2 (Recommendations for the authors):The authors have addressed the points raised in the prior review and made major updates to the manuscript that have improved the clarity of the main findings. In my view, the only area not addressed in depth concerns the summary comment 2, recommending that " an in-depth analysis of the most relevant genes and pathways is necessary". The authors have added some additional detail on gene functions and homologs, however this seems limited, and the authors note that this is in part due to the lack of prior predictive functional information for such developmental genes.

We appreciate the positive comments. Indeed, functional knowledge on Basidiomycota developmental biology is too scanty to perform an in-depth functional characterization of the genes we identified. As a workaround to demonstrate the relevance of developmentally expressed genes to complex multicellularity, we performed a validation analysis (see lines 151-155 and Supplementary File 1). Although this is admittedly circumstantial to the Reviewer’s question, it highlights the scarcity of functionally characterised genes we referred to in our previous answer.

Reviewer #3 (Recommendations for the authors):In this revised manuscript, Merenyi et al. have provided very useful new elements of context and logic to clarify the rationale of their study. I appreciated very much making the list of CM gene candidates explicit and the comments on a number of molecular functions that may be relevant. There has been an effort to connect pieces of the study together that allowed me to get a better view of the overall reasoning and implications of the work. Nevertheless, I believe there is a need to further clarify the overall study objectives and logic, and to avoid overstatements.Indeed, I still find the study objectives fuzzy and the logic somewhat confusing. L77-79 states "The goal of this study was to systematically tease apart the components and driving forces of transcriptome evolution in CM fungi and to identify conserved CM-related genes in the Agaricomycetes", L60-66: "how species-specific and conserved genes, long non-coding RNA, natural antisense transcripts, allele specific expression, small RNA, […] contribute to CM is not known". In the response to reviewers' comments: "one of the novel aspects of our manuscript is the identification of a very specific set of candidate genes". In my view these are incompletely met in this manuscript for the following reasons.

We appreciate pointing out these shortcomings and rephrased several parts of the manuscript to clarify our objectives and logic.

1. The identification of CM-related genes does not include experimental validation (either partial or indirect). The proposed list is therefore speculative, and any evolutionary inference is dependent on the validity of this CM-related gene candidates. I acknowledge validation can be seen as follow up work, in which case I would urge the authors not to overstate the validity of their candidate list.

We toned down our statements on the relationship between the identified phenomena and genes with CM. We also added a literature based validation for developmental genes (see lines 151-155 pdf and Supplementary File 1), which shows that our search for developmentally regulated genes captured known CM-related genes with good sensitivity (but provides no data on specificity).

2. The distinction between CM gene candidates and sexual development genes comes at the end of the Results section with the comparison with C. neoformans, after evolutionary/conservation analyses are presented: While CM-gene candidates are identified in the section L524 and on ("To distinguish genes related to basic sexual processes […] from those related to CM"), some analyses are presumably targeted to CM gene candidates earlier in the manuscript, such as L166 ("To examine how these young genes contribute to the CM transcriptome"), L194 ("young genes […] do not have a uniform contribution to CM development"), L288 ("These examples highlighted the importance of allele specific expression in generating expression variance in CM-related genes"). With an aim to study evolutionary/conservation patterns in CM genes, I would have expected the identification of CM genes to be reported first.

Thank you for drawing our attention on this confusing word usage. We think the root of this inclarity was improper usage of terms in the ms. In the first part of the manuscript we always referred to the developmentally expressed genes of P. ostreatus (or P.gracilis), while in the second part we refer to conserved ortholog groups which show shared sexual-development related or CM-specific expression. To avoid confusion, we clarified the terminology in the first part of the ms.

From the response to previous reviewers' comments, I understand that the TAI pattern and NAT occurrence are not significantly different between CM gene candidates and sexual development genes, while ASE is characteristic for young genes regardless of their putative involvement in CM. Although global patterns are interesting, how gene age, NATs and ASE contribute to CM therefore remains elusive to me.

We added sentences to the manuscript to clarify the connection between gene age, ASE and the developmentally expressed genes.

3. In a number of places, the authors tend to generalize their findings to multiple fungal lineages while some of the analysis only include one or two species (P. ostreatus and Pt. gracilis). For instance, L77 "the components […] of transcriptome evolution in CM fungi" such as NATs and ASE are studied in P. ostreatus and Pt. gracilis and generalization to CM fungi might be premature. L22-25 "We here reveal that allele-specific expression, natural antisense transcripts […] act in concert to shape the transcriptome of complex multicellular fruiting bodies of fungi" is in fact shown for P. ostreatus only. In the title "the transcriptional landscape of sexual morphogenesis" has been studied mostly in P. ostratus and may not generalize to "multicellular fungi". Given the diverse patterns revealed by the TAI analysis encompassing multiple species, whether the ASE and RNA editing patterns are "general principles" (L213) remains speculative.

We changed the title, revised the abstract and the manuscript to avoid unsupported generalizations.

4. The "predictive value" of the study put forward in the title and abstract is confusing to me. e.g. L25 "evolutionary age predicts, to a large extent, the behaviour of a gene in the CM transcriptome", it is not completely clear to me what can be predicted. I am rather used to see the term "predict" used in relation to a formal model the performances of which are quantitatively evaluated. Generally speaking, the connection between the manuscript title, the proposed study objectives and major findings from the work is quite cryptic to me.

We rewrote the abstract and the title, and in the new version of the manuscript we removed the term ‘predict’.